# GenomePAM directs PAM characterization and engineering of CRISPR-Cas nucleases using mammalian genome repeats

Miao Yu [1,2], Limei Ai[1,2], Bang Wang[2,3], Shifeng Lian[2,4], Lawrence Ip[3], James Liu [3], Linxian Li[2,4,5,6], Shengdar Q. Tsai [7], Benjamin P. Kleinstiver [8,9,10] & Zongli Zheng [1,2,4,5,11,12] ✉

Characterizing the protospacer adjacent motif (PAM) requirements of different Cas enzymes is a bottleneck in the discovery of Cas proteins and their engineered variants in mammalian cell contexts. Here, to overcome this challenge and to enable more scalable characterization of PAM preferences, we develop a method named GenomePAM that allows for direct PAM characterization in mammalian cells. GenomePAM leverages genomic repetitive sequences as target sites and does not require protein purification or synthetic oligos. GenomePAM uses a 20-nt protospacer that occurs ~16,942 times in every human diploid cell and is flanked by nearly random sequences. We demonstrate that GenomePAM can accurately characterize the PAM requirement of type II and type V nucleases, including the minimal PAM requirement of the near-PAMless SpRY and extended PAM for CjCas9. Beyond PAM characterization, GenomePAM allows for simultaneous comparison of activities and fidelities among different Cas nucleases on thousands of match and mismatch sites across the genome using a single gRNA and provides insight into the genome-wide chromatin accessibility profiles in different cell types.

In prokaryotes, the CRISPR-Cas system provides antiviral immunity by recognizing and disrupting intruding viral DNA through DNA sequence recognition[1]. This system has been harnessed for precise genome editing in various organisms and cell types[1–5]. Identifying naturally occurring Cas nucleases and engineering Cas enzyme variants with different features is crucial for various research and clinical applications. CRISPR-Cas nucleases form protein:DNA contacts to initiate target site recognition through a protospacer adjacent motif (PAM)[6–11]. The location (5′ or 3′ of the spacer) and sequence of the PAM differs among different types of CRISPR-Cas system[11,12]; however, efficient and accurate identification of PAM requirements in eukaryotic cells remains a bottleneck in the discovery and characterization of novel Cas nucleases and their engineered variants.

Various methods have been developed for PAM identification, including in silico[13] and in vitro cleavage assays[12,14–16], bacterial-based

assays[7,10] including the PAM screen achieved by NOT-gate repression (PAM-SCANR)[17], cell-free transcription–translation (TXTL) systems[18,19], fluorescence-based[20] assays including the PAM definition by observable sequence excision (PAM-DOSE)[21], human cell library-based approaches[22,23] and scalable human cell expression followed by the in vitro cleavage reaction hybrid method, high-throughput PAM determination assay (HT-PAMDA)[24,25]. Each method has its own advantages and limitations (reviewed elsewhere[13,25]). In general, in silico and bacterial depletion results may not be easily translated to mammalian cell contexts. In vitro cleavage assays have the advantages of managing large libraries[15]; however, in vitro methods require laborious protein purification, and the cleavage kinetics may not accurately reflect the conditions in a living organism. Previous in vivo methods require introducing synthetic random oligos as PAM candidates into live cells, which are challenging for maintaining high-diversity sequence libraries.

Moreover, fluorescence (GFP or RFP)-based enrichments are associated with particularly low efficiency.

Repetitive sequences in the mammalian genome, flanked by diverse sequences, are a potential resource for characterizing the PAM preferences of naturally occurring and engineered Cas nucleases. Here we developed a direct PAM identification method called GenomePAM, which uses highly repetitive sequences in the mammalian genome. To characterize the PAM requirements of Cas enzymes, we identified genomic repeats flanked by highly diverse sequences where the constant sequence can be used as the protospacer in CRISPR-Cas genome editing experiments. The cleaved genomic regions can then be analysed using methods such as the genome-wide unbiased identification of double strand breaks (DSBs) enabled by sequencing (GUIDE-seq)[26] that enriches double strand oligodeoxynucleotide (dsODN)-integrated fragments by anchor multiplex PCR sequencing (AMP-seq)[27]. GenomePAM is highly efficient and accurate at characterizing the mammalian-cell-based PAM of an enzyme because every single cell contains just one full set of identical-complexity candidate PAM library. GenomePAM can also simultaneously assess the potency of thousands of on-target sites across the genome and the fidelity of tens of thousands of potential off-target sites of a Cas nuclease, facilitating performance comparison of different Cas nucleases. Moreover, GenomePAM can be used to better understand and compare genome-wide chromatin accessibility profiles of different cell types.

## Results

### Method design

The human genome contains highly repetitive sequences[28], most of which are not suitable for use as protospacers due to low-complexity flanking sequences. However, a subset of these sequences can be used for PAM preference identification, provided they have the following features: (1) The number of unique flanking sequences of a given length in the human genome is comparable with, or not significantly smaller than, the number of potential PAMs to be tested. For example, the PAM of SpCas9 (*Streptococcus pyogenes* Cas9) and its variants may range from 1 to 3 bases; therefore, the number of unique 3-nt-long flanking sequences should preferably be 64 (=$4^3$). In the case of a SaCas9 (ref. 29) (*Staphylococcus aureus* Cas9) and its variants, the PAM may range from 3 to 4 bases; therefore, the number of unique flanking sequences of 4-nt length should preferably be ~256 (=$4^4$). (2) The flanking sequences should have highly diverse or nearly completely random sequence compositions.

To characterize the PAM of SpCas9 and its variants, we analysed the human genome for all possible 20-nt-long sequences and their flanking sequence diversities. For example, there are 8,471 occurrences of the sequence 5′-GTGAGCCACTGTGCCTGGCC-3′ (part of an Alu sequence; hereafter referred to as 'Rep-1') distributed across the human genome (Fig. 1a; ~16,942 occurrences in a human diploid cell) with nearly random flanking sequences of 10-nt length at its 3′ end, making it a suitable candidate as the protospacer sequence for PAM identification (Fig. 1b). For type II Cas nucleases with 3′ PAMs, such as SpCas9 and SaCas9, Rep-1 can be directly used for PAM preference characterization. For type V Cas nucleases with their PAM at the 5′ end of the spacer, such as FnCas12a[12] (*Francisella novicida* Cas12a), the reverse complementary sequence 5′-GGCCAGGCACAGTGGCTCAC-3′ ('Rep-1RC') can be used as the protospacer sequence (Fig. 1b). Since Cas nucleases can often tolerate a few base mismatches (off targets), we calculated the numbers of 20-nt sequences with 1, 2, 3 and 4 base mismatches. For Rep-1, these numbers were 48,207, 206,767, 579,336 and 1,350,488, respectively, and >2 million in total in the human genome (hg38). Thus, using Rep-1 or Rep-1RC as the protospacer, there are potentially >4 million targets in a single human diploid cell. A list of example repeats, their occurrences, flanking sequence diversity and their use are shown in Extended Data Fig. 1.

To leverage these genomic repeats to characterize the PAM of various CRISPR-Cas enzymes, the repeat sequence Rep-1 was chosen as the

protospacer target. The corresponding spacer was cloned into a guide RNA (gRNA) expression cassette to be used along with a plasmid that encodes the candidate Cas nuclease. To identify which repeats within the genome were cleaved in an experiment, we adapted the GUIDE-seq[26] method to capture cleaved genomic sites in HEK293T cells (Fig. 1c). Only those sites whose flanking sequences contain functional PAMs can be cleaved by the Cas nuclease. Cell toxicity after large numbers of DSBs occur in one cell was reported previously when using CRISPR to target highly repetitive element LINE1 (ref. 30) or unique repeat sequences associated with temozolomide mutational signature[31]. To assess this toxicity, we measured cell viability in four different conditions, including Lipofectamine 3000 transfection controls, and in two different cell lines (Methods). The results showed largely similar cell viability across different transfection conditions at 24 h and 48 h after transfection in HEK293T (Extended Data Fig. 2a) and HepG2 cells (Extended Data Fig. 2b). During the GUIDE-seq data analysis, the candidate PAM was set as unknown ('NNNNNNNNNN') and 13,908 sites across the genome were identified (Fig. 1d). The mismatch bases were typically located at positions 8–11 of the targets and were transitions of the intended bases (Fig. 1d). The resulting PAMs were then summarized using their corresponding read counts as weights and used for SeqLogo plotting (Fig. 1e), which was stratified by perfect-match and mismatch targets. Beyond the descriptive SeqLogo, the consistent genomic background sequences inspired us to create an iterative 'seed-extension' method. This approach identifies statistically significant enriched motifs and reports the percentages of edited genomic sites at each iteration step (Fig. 1f and Methods).

### Performance of GenomePAM on SpCas9, SaCas9 and FnCas12a

To evaluate the performance of GenomePAM, we chose three Cas nucleases with well-established PAMs: SpCas9, SaCas9 and FnCas12a. We used Rep-1 for GenomePAM analysis of SpCas9 and SaCas9, and Rep-1RC for FnCas12a. The results showed that the PAM preferences for SpCas9, SaCas9 and FnCas12a were NGG at 3′, NNGRRT (R is G or A) at 3′ and YYN (Y is T or C) at 5′ sides of the spacers, respectively, consistent with previous results[7,10,12,29,32] (Fig. 2a–c; top, perfect match; bottom, mismatch; Fig. 2d–f, 4-base heat map of relative PAM cleavage value (PCV); Methods; GUIDE-seq results are listed in Supplementary Tables 1–3 and visualized in Supplementary Tables 4–6). GenomePAM Table analysis showed that, for SpCas9 3′ PAM, the most significant single base was the G at position 3 (1,103 (65.6%) of total 1,681 targets in human genome edited), the most significant two bases were GG at positions 2 and 3 (449 (94.1%) out of total 477 targets edited), and no further significant bases (Fig. 2g). For SaCas9 3′ PAM, the GenomePAM Table analysis showed increasing significance for G at position 3, GR at 3–4, GRR at 3–5, and GRRT at 3–6, respectively (Fig. 2h). The corresponding percentages of edited targets were 44.6%, 62.1–73.9%, ~80.0–93.3%, and 96.7–98.0%, respectively (Fig. 2i). For FnCas12a 5′ PAM, the GenomePAM Table analysis showed a Y position −3 (5.3–6.0% of targets edited) and a YY at position −2 to −3 (8.5–9.6% of targets edited). For the known PAM wobble bases[6,33], namely the 2nd position of N[G/A]G in SpCas9 and the 6th position of NNGRR[T/A] in SaCas9, the proportions of the prominent bases ([G] in SpCas9 and [T] in SaCas9) in the mismatch target-associated PAMs were higher than those perfect-match-associated PAMs (Fig. 2j; left, SpCas9; right, SaCas9; both $P < 0.01$). We also performed validation experiments in three other cell lines, HepG2, Huh7 and HeLa cell lines (Extended Data Fig. 3). The results showed nearly identical PAM profiles as those using HEK293T cells. Furthermore, we performed further tests using another repeat sequence (Rep-2: 5′-GAGCCACCGTGCCTGGCCTC-3′) that occurs 1,126 times in the human genome (~2,252 occurrences in a human diploid cell) (Extended Data Fig. 1) as the protospacer for GenomePAM analysis of SpCas9, SaCas9 and FnCas12a. The GenomePAM results were nearly the same, namely, NGG at 3′, NNGRRT at 3′ and TTTN at 5′ for the three nucleases, respectively (Extended Data Fig. 4). We further

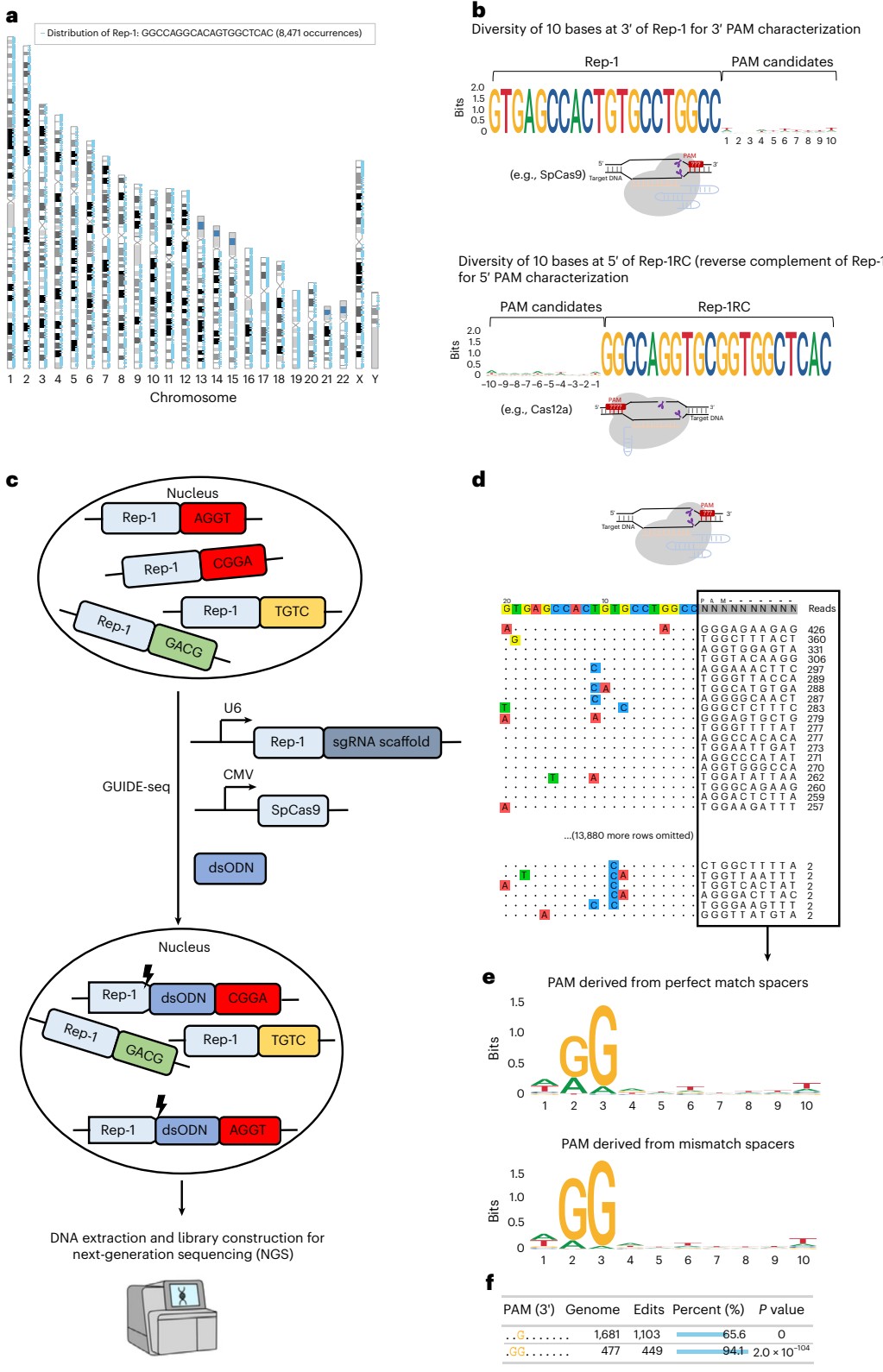

**Fig. 1 | Method design. a**, Genome-wide distribution of the Rep-1 sequence in the human genome. **b**, SeqLogo plot showing nucleotide frequency at each position in the 10 bases at 3′ of Rep-1 and the 10 bases at 5′ of Rep-1RC (reverse complement of Rep-1). Rep-1 and Rep-1RC sequences can be used to characterize PAM preferences of Cas nucleases with 3′ PAM and 5′ PAM, respectively. **c**, GenomePAM workflow for the identification of PAM preferences using the highly repetitive genome sequence Rep-1 as the protospacer and the GUIDE-seq experiments to capture cleaved genomic sites of SpCas9. CMV, cytomegalovirus promoter. **d**, An example of GUIDE-seq output. Each line shows one SpCas9

cleavage site and the mismatch bases are colour coded. The flanking sequences and GUIDE-seq read counts of each site are shown on the right side. **e**, SeqLogo plot summary for SpCas9 PAM preferences using their corresponding read counts as weights and stratified by perfect-match and mismatch targets. **f**, GenomePAM Table reporting the enriched PAM sequences and counts, along with the numbers and percentages of corresponding genomic sites edited, and associated statistical significance. *P* values were derived from two-sided chi-square test.

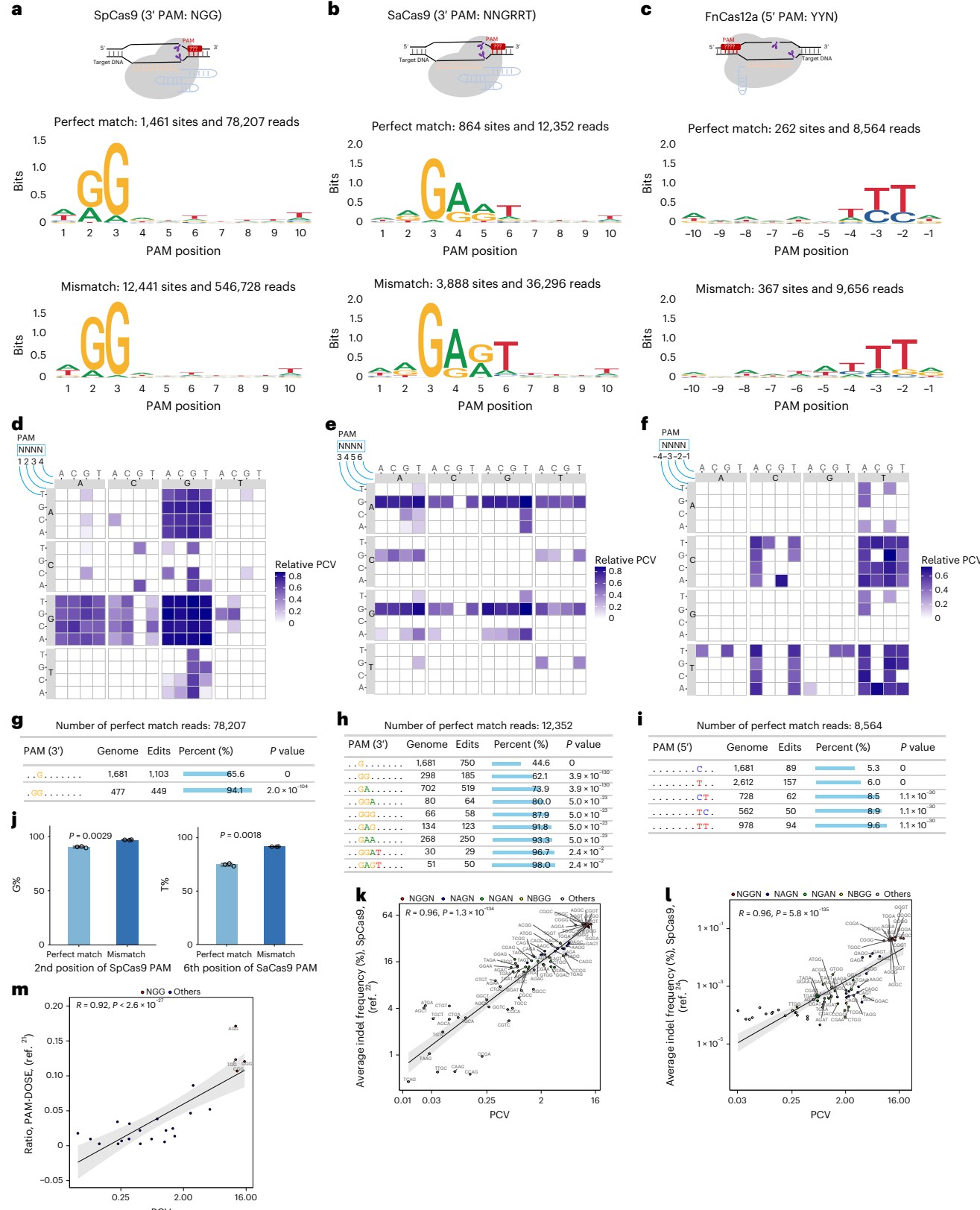

**Fig. 2 | Evaluation of the GenomePAM assay on the PAM characterization for SpCas9, SaCas9 and FnCas12a. a–i,** SeqLogo results for SpCas9, SaCas9 and FnCas12a PAM preferences in HEK293T cells with perfect-match spacers (**a–c**, top) and with mismatch spacers (**a–c**, bottom), and plotted in a 4-base heat map of relative PCV (**d–f**) and corresponding GenomePAM Tables (**g–i**). *P* values were derived from two-sided chi-square test. **j,** Percentage of 'G' at the 2nd position of SpCas9 PAM (left) and percentage of 'T' at the 6th position of SaCas9 PAM (right), by cleaved target types (perfect match versus mismatch). Data are presented as mean ± s.d. **k–m,** Correlations between PCV of GenomePAM and indel frequencies reported by previous methods: ref. 22 (**k**), HT-PAMDA[24] (**l**) and PAM-DOSE[21] (**m**) across various SpCas9 PAM sequences, including canonical (NGGN) and non-canonical PAMs (NAGN, NGAN, NBGG; 'B' is C, G or T). Linear regression lines are plotted, with 95% confidence intervals indicated as grey areas.

compared SpCas9 canonical (NGGN) and non-canonical PAM (NAGN, NGAN, NBGG; 'B' is C, G or T) PCVs derived from GenomePAM with indel frequencies reported using three well-established methods[21,22,24]. The analyses showed high correlations (Fig. 2k,l; versus two assays: $R = 0.96$, $P < 1 \times 10^{-100}$; Fig. 2m, versus PAM-DOSE: $R = 0.92$, $P < 2.6 \times 10^{-27}$). Our results recapitulate the known PAM requirements of type II and type V Cas nucleases, demonstrating that our genome-based PAM determination method (GenomePAM) is effective.

## Characterization of challenging PAM preferences

Long and complicated PAMs in naturally occurring Cas nucleases pose a challenge to identifying their PAMs experimentally. We evaluated the performance of GenomePAM on *Campylobacter jejuni* Cas9 (CjCas9), which was reported to require a 7-nt PAM NNNNACA[34] and an 8-nt one NNNNRYAC[35]. We performed GenomePAM assay on CjCas9 using Rep-1 as the protospacer in HEK293T cells. The results showed that CjCas9 required NNNNRYAC as its PAM in HEK293T cells (Fig. 3a; top, perfect match; bottom, mismatch). Because the optimal length of protospacer for CjCas9 was shown to be 22 bases[35], we tested using extended Rep-1 to 21 (5′-YGTGAGCCACTGTGCCTGGCC-3′; Y is C or T) and 22 (5′-GYGTGAGCCACTGTGCCTGGCC-3′) bases. The results showed nearly the same NNNNRYAC PAM preferences for both 21 and 22 base protospacers (Fig. 3b,c; top, perfect match; bottom, mismatch). Relative PCVs using these protospacers are visualized in heat maps (Fig. 3d–f). GenomePAM Table analysis consistently showed the most enriched sequence ACAC at positions 5–8 (Fig. 3g–i). The longer protospacers with 21 and 22 bases showed marked increases in cleavage activities compared with the 20-base spacer for CjCas9 (Fig. 3j). The PAM requirement on the 8th position was relatively relaxed (Fig. 3b,c), and there were increases in the numbers of off-target sites (Fig. 3a–c, bottom).

Another challenging scenario in characterizing PAM preference is when there is little preference. Engineering Cas nucleases to relax PAM requirements can broaden potential applications, such as using the SpRY variant of SpCas9 with nearly no PAM restriction (previously described to be NRN > NYN)[24]. Additional near-PAMless Cas variants have also been developed[36,37]. However, depletion-based methods may not be efficient in identifying Cas nucleases with nearly no PAM preferences[25]. Being a positive selection method, GenomePAM found that as expected, SpRY exhibited a very minimal PAM requirement, being nearly PAMless across 5,003 perfect-match loci and 23,946 mismatch loci in HEK293T cells (Fig. 3k).

## GenomePAM for characterizing novel Cas PAM

After establishing the simplicity and accuracy of the GenomePAM assay in identifying various Cas nucleases, we sought to demonstrate its utility in PAM identification for novel Cas discovery. Using a metagenomics approach (Methods) to analyse recent data in the NCBI Sequence Read Archive (SRA), we identified one novel type V-A CRISPR–Cas candidate derived from *Ruminococcus Dsp902787825*, named RuCas12a (Fig. 4a). We performed the GenomePAM assay using 'Rep-1' as the protospacer to characterize its potential activity and 5′ PAM requirement. The result revealed that the PAM preference of RuCas12a was TTYN at its 5′ end (Fig. 4b,c). The GenomePAM Table showed dominant TTC at positions −4 to −2 (Fig. 4d, bottom row). To further validate and comprehensively evaluate genome editing efficiency of RuCas12a, we used 20 regular (non-repetitive) genomic sites containing a 5′ end 'TTTG' in human genes *CD34*, *CFTR*, *DNMT1*, *EMX1*, *HBB*, *LPA*, *POLQ*, *RFN2*, *TTR* and *VEGFA* (spacer and primer sequences are listed in Supplementary Table 7). The editing efficiencies ranged from 3.4% to 40.6% across the 20 genomic loci in HEK293T cells (Fig. 4e). We also applied GenomePAM for identifying PAM of novel type II Cas nuclease and found a novel Cas9 from *Tissierella* sp., named TiCas9. TiCas9 clusters closely to SpCas9 and ScCas9, implying that it is a type II-A Cas nuclease (Fig. 4f). GenomePAM analysis revealed that TiCas9 had an NNNACT PAM (Fig. 4g–i).

We further validated its potencies across 20 endogenous loci with a 3′-NNNACT PAM in genes *CD34*, *CTCF*, *EMX1*, *POLQ* and *VEGFA* in HEK293T cells, which showed up to ~30% editing efficiency using its native gRNA scaffold (Fig. 4j; spacer and primer sequences are listed in Supplementary Table 8).

## GenomePAM facilitates Cas PAM engineering

Engineering Cas PAM preference to expand targetability represents an attractive strategy for broad applications[37]. To this end, we questioned whether GenomePAM could facilitate Cas variant discovery. We assessed this using TiCas9 as an example by first applying Genome-PAM to profile pooled mutant variants and, upon evidence of altered mixed PAMs, applied GenomePAM characterization of single-mutant variants (Fig. 5). Because there are many Cas9 nucleases recognizing G/C-rich PAM, we aimed to engineer TiCas9 for recognizing A/T-rich PAM, namely, to relax the C at position 5 of NNNACT. Using AlphaFold 3 (ref. 38), we identified that K1315 was the only residue found to interact with G at position 5 on the complementary strand (Fig. 5a). We constructed an NNK library encoding for all 20 amino acids at position 1315. GenomePAM analysis of the pooled variants showed dramatically altered base compositions at position 5, without affecting positions 4 and 6, in the aggregated PAMs (Fig. 5b–d). We then assessed all the 19 a.a. variants individually. The variant K1315Q showed completely no restriction at position 5 (Fig. 5e–g), namely, an ANT PAM at positions 4–6, while another 18 variants showed varied preferences at position 5 (Extended Data Fig. 5). Then, 16 endogenous sites in *RNF2* harbouring PAM positions 4–6 (4 ACT, 4 ATT, 4 AGT and 4 AAT) were used to validate the variant K1315Q versus wild type (WT) (spacer and primer sequences are listed in Supplementary Table 9). The results were consistent with SeqLogo, PCV visualization and the GenomePAM Table (Fig. 5h versus 5b–g). Interestingly, even though the SeqLogo of the variant K1315Q showed no noticeable dominant base at position 5 (Fig. 5e), the GenomePAM Table (Fig. 5g) showed that, for PAM positions 4–6, the proportions of genome-wide target sites edited were highest with AGT, followed by ACT, and the lowest with AAT and ATT, largely consistent with the indel percentages at the 16 endogenous sites tested individually (Fig. 5h).

## Comparison of genome-wide potency and specificity

Many SpCas9 variants have been developed to reduce off-target effects (for example, SpCas9-HF1 (ref. 39), HypaCas9 (ref. 40), eSpCas9(1.1)[41], Sniper-Cas9 (ref. 42) and Sniper2L-Cas9 (ref. 43)) and broaden PAM compatibilities (for example, xCas9 (ref. 44)). Oftentimes, dozens of gene loci are used to assess the fidelity and activity of Cas9 variants[40,42]. It would be desirable to simultaneously evaluate Cas nuclease potency and specificity with a less laborious method than traditional library-based approaches[22,45]. A method based on large-scale synthetic oligos ($n = 26,891$) containing targeting sequences and mismatch sequences has been developed for this purpose[22]. GenomePAM uses a single protospacer oligo that provides thousands of perfect-match sites and millions of mismatch sites in one human cell. Therefore, we sought to evaluate the feasibility of simultaneously comparing genome-wide potency and specificity of different Cas9 variants. We performed GenomePAM experiments with 'Rep-1' as the targeting protospacer for WT SpCas9 and six variants in parallel (SpCas9-HF1, eSpCas9(1.1), HypaCas9, xCas9, Sniper-SpCas9, and Sniper2L-SpCas9), with the same amounts of Cas and of sgRNA expression plasmids (Extended Data Fig. 6). The ratios of on-to-off target sites were highest for SpCas9-HF1 (mean 1.13), eSpCas9(1.1) (mean 1.08) and HypaCas9 (mean 0.93), followed by xCas9 (mean 0.57), Sniper2L-SpCas9 (mean 0.23) and Sniper-Cas9 (mean 0.20), and lowest for WT SpCas9 (mean 0.13) (Fig. 6a). Similarly, the ratios of on-to-off target reads were 1.12, 2.28, 0.78, 0.75, 0.36, 0.31 and 0.17, respectively (Fig. 6b). To evaluate Cas9 cleavage dynamics with increasing probing data, we sampled datasets from 100,000 up to 1 million sequencing reads for GUIDE-seq analysis.

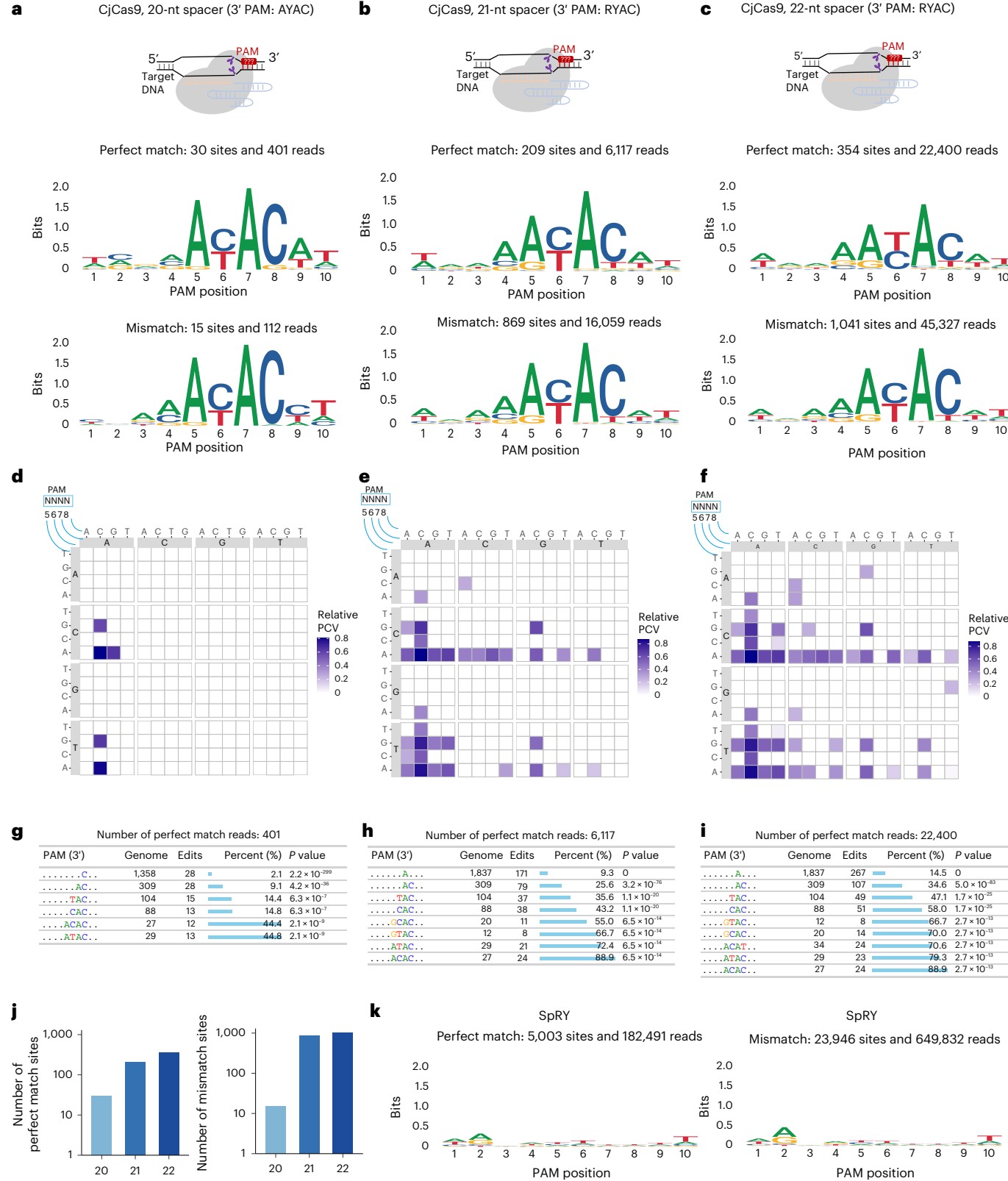

**Fig. 3 | Evaluation of the GenomePAM assay on the PAM characterization for CjCas9 and SpRY. a–i**, SeqLogo results for CjCas9 when using different lengths of spacers: 20 bases ('GTGAGCCACTGTGCCTGGCC') (**a**), 21 bases ('YGTGAGCCACTGTGCCTGGCC'; 'Y' is 'C' or 'T') (**b**) and 22 bases ('GYGTGAGCCACTGTGCCTGGCC') (**c**) with perfect-match cleaved sites (top) in HEK293T cells and in their mismatch cleaved sites (bottom), and corresponding 4-base heat map of relative PCVs (**d–f**) and GenomePAM Tables (**g–i**). *P* values were derived from two-sided chi-square test. **j**, The numbers of perfect-match sites (left) and mismatch sites (right) cleaved by CjCas9 using 20-nt, 21-nt and 22-nt spacers. **k**, SeqLogo results for SpRY PAM preferences in HEK293T cells with perfect-match spacers (left) and mismatch spacers (right).

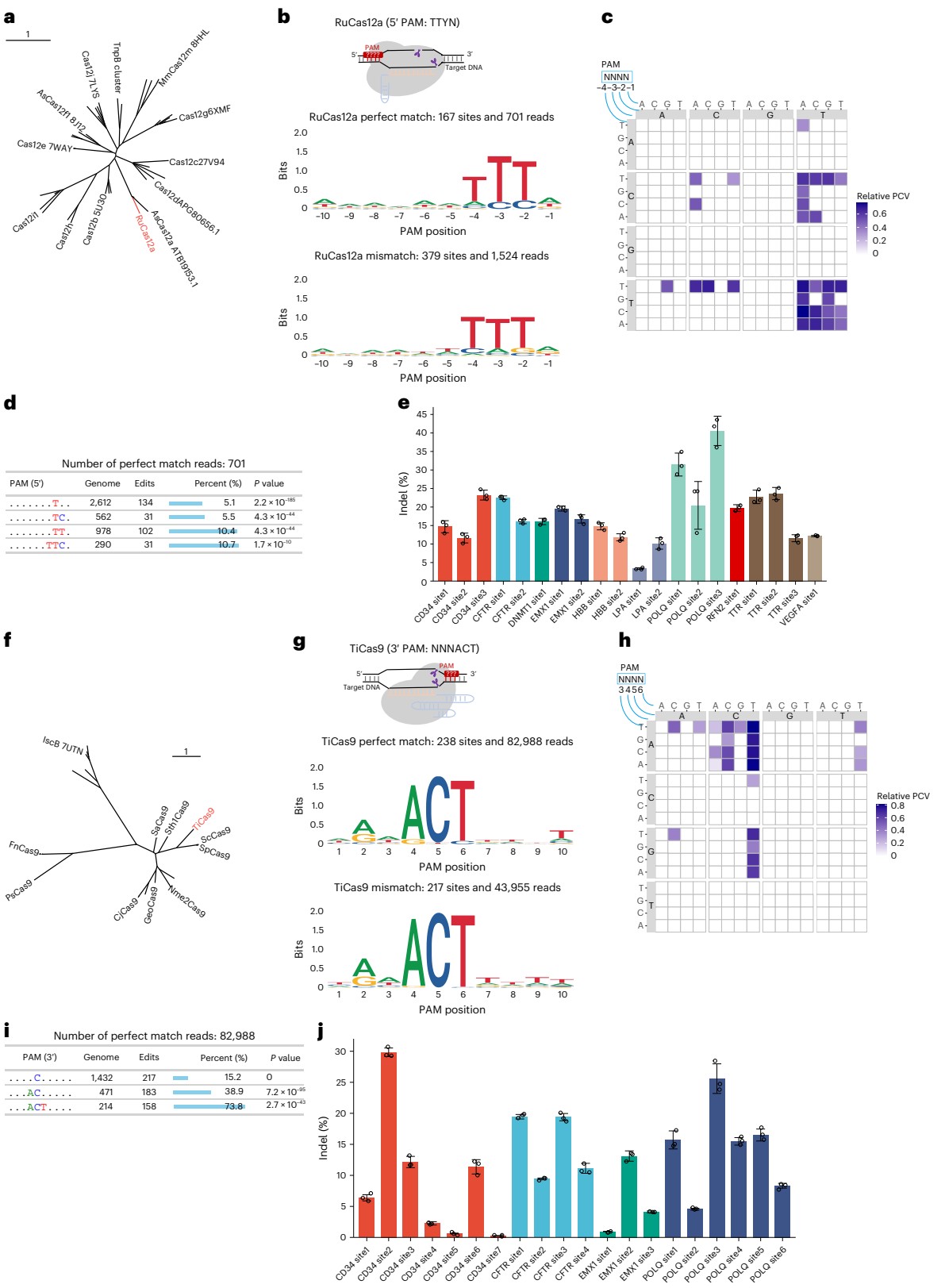

**Fig. 4 | Novel Cas nuclease discovery and their PAM identifications using the GenomePAM assay. a**, Phylogenetic tree for the type V-A Cas system, RuCas12a. **b**–**d**, SeqLogo results for RuCas12a PAM preferences in HEK293T cells with perfect-match spacers (**b**, top) and mismatch spacers (**b**, bottom), associated 4-base heat map of relative PCV (**c**) and the GenomePAM Table (**d**). *P* values were derived from two-sided chi-square test. **e**, Editing efficiency of RuCas12a on 20 regular genomic sites in human genes *CD34*, *CFTR*, *DNMT1*, *EMX1*, *HBB*, *LPA*, *POLQ*, *RFN2*, *TTR* and *VEGFA* with a 5′-TTTG PAM. Data are presented as

mean ± s.d. **f**, Phylogenetic tree for the type II CRISPR-Cas system, TiCas9. **g**–**i**, SeqLogo results for TiCas9 PAM preferences in HEK293T cells with perfect-match spacers (**g**, top) and mismatch spacers (**g**, bottom), and a 4-base heat map of relative PCV (**h**) and the GenomePAM Table (**i**). *P* values were derived from two-sided chi-square test. **j**, Editing efficiency of TiCas9 on 20 regular genomic sites in human genes *CD34*, *CTCF*, *EMX1*, *POLQ* and *VEGFA* with a 3′-NNNACT PAM. Data are presented as mean ± s.d.

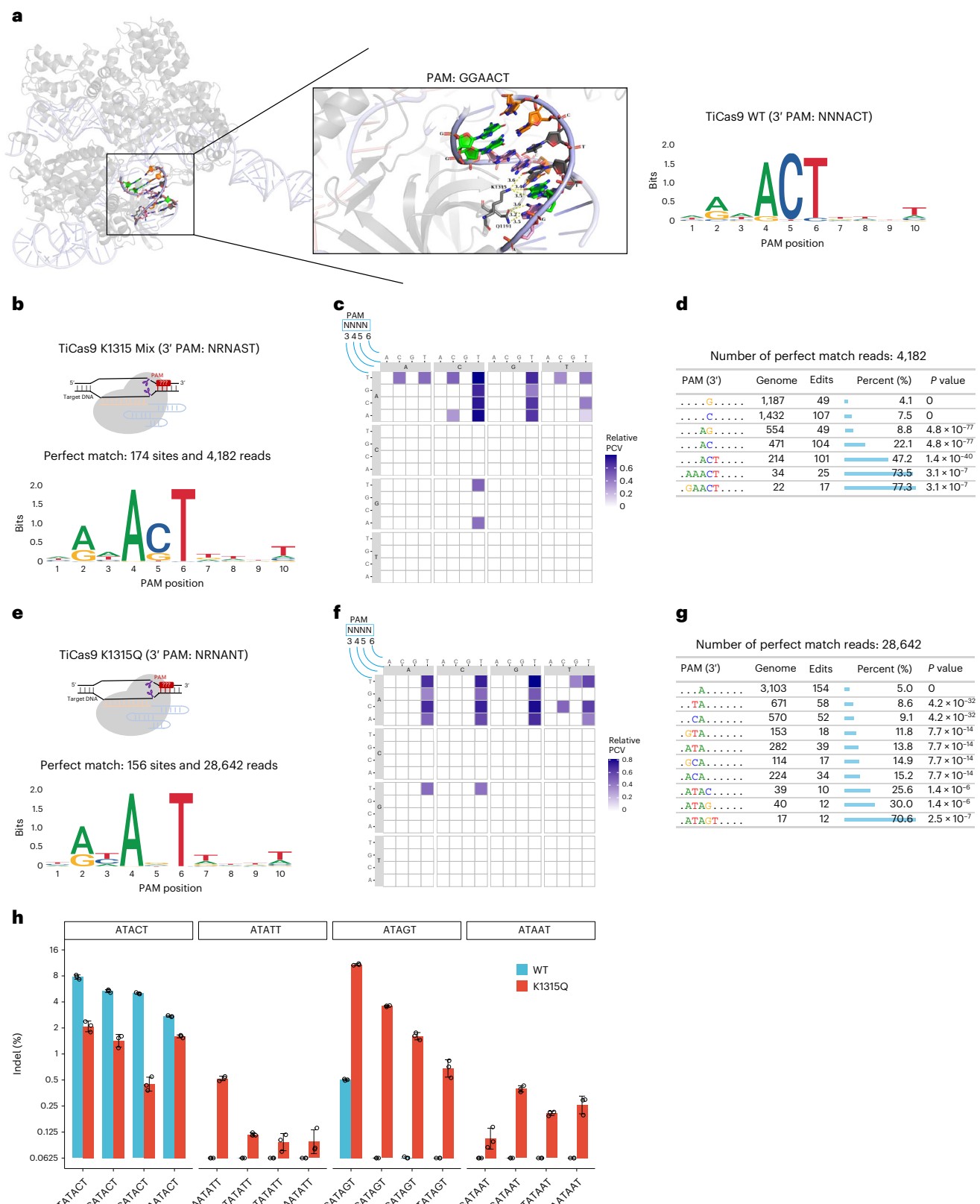

**Fig. 5 | Structure-guided engineering for altered PAM preference of TiCas9.**
**a**, Left: the structure of the TiCas9 and sgRNA complex was predicted using AlphaFold 3 and visualized in PyMOL 3.0. The PAM-interacting domain containing polar residue–DNA interactions within 4 Å are shown. Middle: hydrogen bonds between K1315 or Q1191 and nitrogenous base in the anti-sense strand of PAM are indicated by yellow dashed line, with corresponding distances labelled. Right: SeqLogo showing the GenomePAM result of TiCas9 WT.
**b**–**d**, SeqLogo visualization of the 3′ PAM preference of the pooled NNK library

of the TiCas9 1315 variants (**b**), the associated 4-base heat map of relative PCV (**c**) and the GenomePAM Table (**d**). *P* values were derived from two-sided chi-square test. **e**–**g**, SeqLogo visualization of the 3′ PAM preference of the TiCas9 K1315Q variant (**e**), the associated 4-base heat map of relative PCV (**f**) and the GenomePAM Table (**g**). *P* values were derived from two-sided chi-square test. **h**, Indel percentages using TiCas9 WT and its K1315Q variant on 16 endogenous sites in *RNF2* harbouring different PAM sequences (positions 4–6: A<u>C</u>T, A<u>T</u>T, A<u>G</u>T and A<u>A</u>T). Data are presented as mean ± s.d.

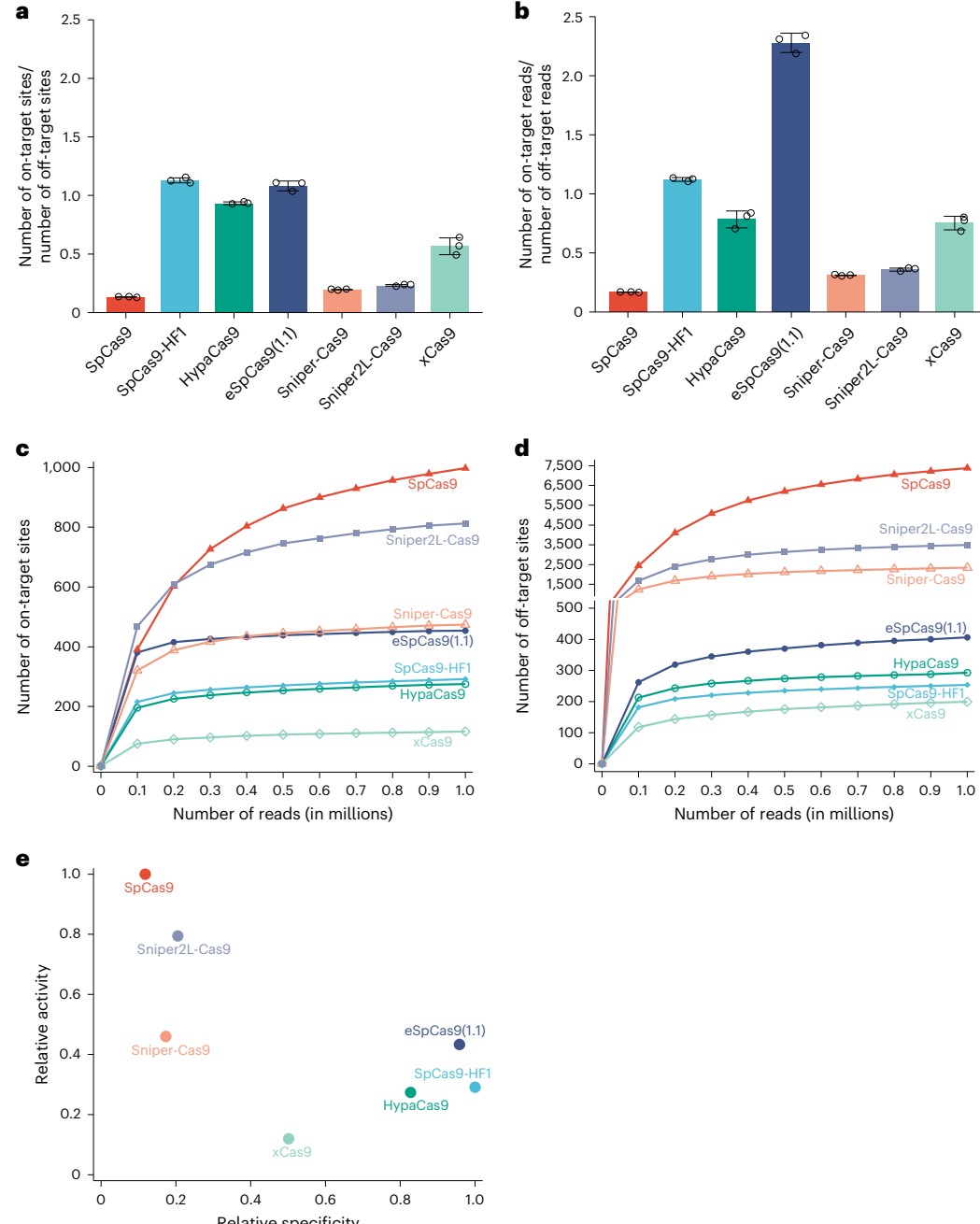

**Fig. 6 | General activities and specificities of different SpCas9 variants.**
**a**, Ratio of the number of on-target sites to the number of off-target sites for seven SpCas9 variants. Data are presented as mean ± s.d. **b**, Ratio of the number of on-target sequencing reads to the number of off-target sequencing reads for seven SpCas9 variants. Data are presented as mean ± s.d. **c,d**, The number of on-target

sites (**c**) and the number of off-target sites (**d**) detected when using randomly downsampled datasets, from 0.1 million to 1 million raw sequencing reads. **e**, Relative activities (defined as the number of perfect-match sites relative to that of WT SpCas9) and specificities (defined as the ratio of perfect-match to mismatch site numbers relative to the ratio in SpCas9-HF1) for seven SpCas9 variants.

The number of on-target sites identified given the same amount of sequencing data was highest (the most potent) in WT, followed by Sniper2L-SpCas9, comparable in Sniper-SpCas9 and eSpCas9(1.1), and lowest in SpCas9-HF1, HypaCas9 and xCas9 (Fig. 6c). The numbers of off-target sites identified given the same amount of data were lowest (the most specific) in xCas9, HypaCas9, SpCas9-HF1 and eSpCas9(1.1), comparable in Sniper-SpCas9 and Sniper2L-SpCas9, and highest (the least specific) in WT (Fig. 6d).

To compare general activity and specificity of different SpCas9 variants in one place, we used the 1M-read datasets. We defined relative activity as the number of perfect-match sites relative to the number of

perfect-match sites identified by WT SpCas9, and relative specificity as the ratio of perfect-match to mismatch target site numbers relative to the same ratio in SpCas9-HF1 (because SpCas9-HF1 had the highest ratio among the seven SpCas9 tested here) (Fig. 6e) The scatterplot showed that WT and Sniper2L-SpCas9s were more potent but less specific than other variants, whereas eSpCas9, SpCas9-HF1 and HypaCas9 were more specific but less potent than WT SpCas9 (Fig. 6e). To evaluate whether using a different repetitive spacer for GenomePAM can affect general activity and specificity, we used Rep-3 (Extended Data Fig. 1) and the results were similar to those obtained with Rep-1 (Extended Data Figs. 7 and 8).

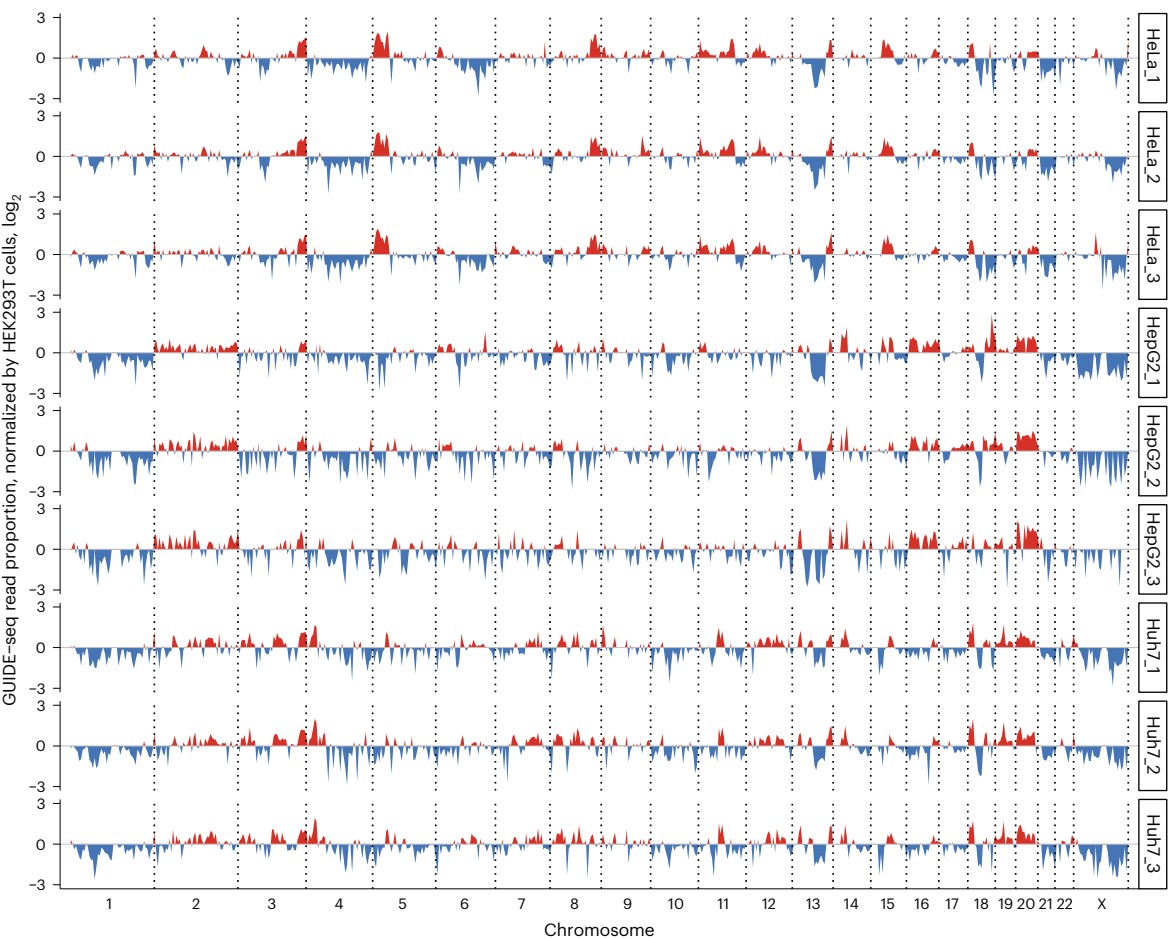

**Fig. 7 | The GenomePAM assay profiles of chromosome accessibility when using SpCas9 and targeting 'Rep-1' in human cell lines HeLa, HepG2 and Huh7.** The proportion of cleavage read counts in each 5-million-base chromosomal window was divided by the proportion in the corresponding chromosomal window in the HEK293T cells (mean of triplicates) and log$_2$ transformed. A higher cleavage proportion relative to HEK293T is coloured in red and a relatively lower proportion in blue.

## Profiling chromatin accessibility in different human cell lines

Chromatin conformation has been shown to affect Cas nuclease genome editing[46,47] on relatively small numbers of genomic targets but not on the genome-wide scale. To assess genome-wide targeting profiles in different cell lines, we performed GenomePAM assays using SpCas9 and 'Rep-1' in HEK293T, HepG2, Huh7 and HeLa cell lines in triplicates (Fig. 7). Genome-wide chromatin accessibility was defined as the number of targeting reads per 5-M-base chromosome window. Relative to HEK293T, the results from triplicates of the same cell line showed consistent and reproducible genome-wide chromatin profiles (Extended Data Fig. 9). Interestingly, hepatocyte-derived cell lines HepG2 and Huh7 showed very similar chromatin accessibility profiles in contrast to HeLa cells, indicating similar chromatin accessibility in the same tissue (Fig. 7).

## Discussion

In this study, we developed a new method called GenomePAM and demonstrated its simplicity, accuracy and capability in assaying PAM preferences of previously established SpCas9, SaCas9 and FnCas12a nucleases, as well as complicated and challenging PAM recognitions as in SpRY and CjCas9. We also demonstrated the potential of GenomePAM for simultaneous comparison of potencies (thousands of perfectly matched loci) and, when combined with GUIDE-seq, fidelities (tens of thousands of off-target sites) of various Cas nucleases and variants. Compared with other PAM identification methods using regular PCR amplicon sequencing, GenomePAM uses GUIDE-seq and thus requires relatively more skills

to perform. However, GUIDE-seq has been one of the main methods for assessing CRISPR off-target effects in both research and therapeutic settings[48,49]. Using GenomePAM, we rapidly identified one type II Cas TiCas9 and one type V Cas RuCas12a that are both active in human cells. Directed by AlphaFold 3, we further accelerated the PAM engineering of TiCas9 to expand its targetability using GenomePAM. In addition, we demonstrated at the genome-scale that genomic accessibility of a given CRISPR-Cas design differs among cells of different tissue types. We envision that GenomePAM will be widely useful for the discovery, characterization and comprehensive evaluation of PAM recognition, potency and fidelity of CRISPR-Cas nucleases and engineered variants.

GenomePAM is capable of direct identification of challenging PAMs in human cells. Different PAM preferences for CjCas9 have been reported. An in silico prediction followed by a biochemical digestion assay reported that the PAM for CjCas9 was NNNNACA[34], while an in vitro cleavage assay followed by in vivo second-step analyses on each of the positions from 5 to 8 showed that the optimal PAM was NNNN-RYAC[35]. Maintaining large-scale libraries consisting of many sequences is challenging. Previous methods have attempted to address this by using sequential rounds of experiments in exceptionally challenging situations, with progressively lengthened PAM candidate sequences[12,14]. Maintaining rich library complexities in large-scale screening experiments is often challenging, but is not an issue in GenomePAM since every single cell contains one full set of potential PAM candidates, and maintaining PAM candidate sequence diversity is also not an issue in GenomePAM. We used GenomePAM in a one-round experiment directly

in human cells and showed that, without previous protein purification and without introducing a library of synthetic oligos, the PAM preference of CjCas9 was NNNNRYAC. Since GenomePAM is a positive selection method, it can be used to efficiently identify PAM requirements when there are no preferences[25].

Methods that can compare potency and fidelity of various Cas nucleases simultaneously are highly desirable. One such method involves constructing stable cell lines with balanced expression of Cas nucleases and variants to be compared, followed by transduction of a large pool of synthetic oligos ($n$ = 26,891, on- and off-target sequences) at a carefully controlled multiplicity of infection (MOI) into these stable expression cells to compare potency and fidelity of various Cas nucleases[22]. One advantage of this approach is that it includes different on-target and off-target sequences. GenomePAM takes advantage of highly repetitive sequences in every cell (thousands of on-target and tens of thousands of off-target sequences), is much simpler and of low cost. One limitation of GenomePAM is that it uses relatively limited kinds of on-target sequences, although they appear thousands of times in one cell. However, this can be compensated for by using different repetitive sequences, such as Rep-1, Rep-3 or Rep-4 for result confirmation and validation, and in different cell types. Indeed, our results showed that using different sequences (Rep-1, Rep-2 or Rep-3) as the protospacers for GenomePAM analyses gave the same results in PAM characterizations for the different Cas nucleases tested in this study. However, a new Cas nuclease might have a scaffold sequence that interferes with the repeats, potentially forming strong secondary structures and affecting GenomePAM results. We recommend using at least two different repeats as GenomePAM spacers for novel Cas nucleases. Another possibility is to combine different repetitive sequences in one experiment, although we have not tested this ourselves yet. In such a case, bioinformatic analysis would need to use one repetitive sequence at a time and repeat the data analysis for all sequences. Chromatin accessibility affects Cas nuclease activity, as shown on a genome-wide scale. The GenomePAM assay is minimally biased by chromatin accessibility, probably due to the large number of accessible perfect-match targets in each cell.

Measuring chromatin accessibility is important in understanding basic cellular processes, including transcription, replication, chromosome segregation and DNA repair[50]. A variety of techniques such as Dnase-seq[51] and ATAC-seq[52,53] enable quantifying genome-wide chromatin accessibility. Genome accessibility to CRISPR-Cas targeting is known to differ among different cell types but has been demonstrated only in limited and selected loci. GenomePAM demonstrates clearly that the CRISPR-Cas genome-wide accessibility profiles differ among different cell types. GenomePAM may complement existing methods for studying genome-wide chromatin dynamics.

## Methods

### Identification of repeat sequences
The human genome (hg38) was used to calculate the frequencies of all 20-mer sequences using jellyfish tools[54]. Because a spacer starting with a G base at 5′ is required for most Cas nucleases, we selected all 20-mer sequences starting with a 5′ G. To avoid simple homopolymers and to increase base composition diversity, we also excluded those 20-mers containing 'AAA', 'TTT', 'CCC' or 'GGG'. Among the remaining 20-mer sequences, to retrieve their flanking sequences, we used BWA[55] to map their chromosomal coordinates, and retrieved 10 bases upstream and 10 bases downstream using samtools[56]. The diversity of the flanking sequences of each of the 20-mer sequence was plotted using the ggseqlogo[57] package. We defined PCV as the ratio of the percentage of a given PAM sequence among all sequences of the same length captured by GenomePAM to the percentage of the same PAM sequence among all sequences of the same length in the human genome (hg38). To better visualize the PAM recognition pattern in a 4-base heat map, a relative PCV was calculated by $\log_2$ transforming PCV and normalizing to the PAM sequence with the highest PCV.

### CRISPR-Cas identification
Metagenomes were downloaded from EMBL-EBI MGnify, NCBI GenBank and the Joint Genome Institute, or assembled in-house using raw sequencing reads from the NCBI Sequence Read Archive. We used a combinatorial pipeline that includes CCTyper[58], CRISPRcasIdentifier[59] and OPFI[60] to predict putative Cas proteins. MinCED[61] was used to identify CRISPR arrays; any CRISPR arrays located adjacent to the predicted Cas, which typically comprises a CRISPR operon, were subjected to further analysis. Putative sequences of Cas were scanned using Interproscan[62] to identify and annotate conserved domains. Selected Cas proteins were aligned with MAFFT[63], and a phylogenetic tree was constructed using FastTree2 (ref. [64]).

### Cell culture
HEK293T (CRL3216, ATCC), HepG2 (CRL11997, ATCC) and HeLa (CCL-2, ATCC) cell lines were purchased from the American Type Culture Collection (ATCC). The Huh7 (01042712, Sigma) cell line was purchased from Sigma. HEK293T cells, HeLa cells and Huh7 cells were cultured in Dulbecco's modified Eagle medium (C11995500BT, GIBCO), HepG2 cells were cultured in Eagle's minimum essential medium (30-2003, ATCC) supplemented with 10% fetal bovine serum (10270-106, GIBCO), and all cells were incubated at 37 °C with 5% $CO_2$ in a constant-temperature incubator. Cell passaging was performed at a 1:3 split ratio when the cells reached 90% confluence.

### Plasmids and oligonucleotides
The plasmids used in these experiments were purchased from the non-profit plasmid repository Addgene. The plasmid lentiCRISPRv2 (Addgene, 52961) was used to express wild-type SpCas9; BPK2139 (Addgene, 65776) to express wild-type SaCas9; pY004 (Addgene, 69976) to express wild-type FnCas12a; pET-CjCas9 (Addgene, 89754) to express wild-type CjCas9; and the plasmids BPK1520 (Addgene, 65777), BPK2660 (Addgene, 70709), pU6-Fn-crRNA (Addgene, 78958) and pU6-cj-E sgRNA (Addgene, 169915) were used to express SpCas9 sgRNA, SaCas9 sgRNA, FnCas12a crRNA and CjCas9 sgRNA, respectively. Oligonucleotide duplexes corresponding to the target spacer sequences were purchased from GENEWIZ.

### Plasmid construction
Oligonucleotide duplexes corresponding to sgRNA sequences (paired top and bottom single-stranded oligos) were annealed together using the following programme: 95 °C, 3 min; 70 cycles of (95 °C, 1 min, with −1 °C per cycle); 4 °C hold. Annealed DNA segments were inserted into BsmbI digested sgRNA expression plasmids. After transformation into bacteria and selection, the plasmids were purified by PureLink HiPure Plasmid Midiprep kit (Invitrogen). Sequences of guide insertion in the plasmids were confirmed by Sanger sequencing (BGI)

### dsODN preparation
dsODN oligos were purchased from GENEWIZ with HPLC purification. Each oligo was resuspended in 1× TE buffer (ThermoFisher, 12090015) to a final concentration of 250 μM. These oligos were then annealed at 100 μM in 1× annealing buffer (10 mM Tris-HCl, 50 mM NaCl, 1 mM EDTA, pH7.4) on a thermocycler. The programme was 95 °C, 3 min; 70× (95 °C, 1 min, −1 °C per cycle); 4 °C hold.

### Cell transfection
Guide RNAs and Cas protein plasmids were transfected into cells using Lipofectamine 3000 transfection reagent (ThermoFisher, L3000015) following manufacturer instructions. Cells were cultured at a density of $1 \times 10^5$ per well in a 24-well plate. For each well, 100 ng of gRNA and 400 ng of Cas expression plasmids together with 5 pmol of annealed dsODN were mixed with 1 μl of P3000 reagents in 25 μl Opti-MEM medium and then mixed with 1.5 μl Lipofectamine 3000 reagent in 25 μl Opti-MEM medium to generate a total volume of 50 μl DNA–lipid

complex, followed by incubation for 10 min at room temperature. The transfection complex was added into individual wells. The plate was maintained in a cell culture incubator for 48–72 h.

## Cell viability assay

HEK293T and HepG2 cells were seeded in 96-well plates at $2.0 \times 10^4$ per well and transfected at four different conditions: (1) SpCas9 plasmid + Rep-1 sgRNA plasmid + dsODN; (2) SpCas9 plasmid + Rep-1 sgRNA plasmid; (3) SpCas9 plasmid + non-targeting sgRNA plasmid + dsODN; and (4) Lipofectamine 3000 only. Cell viability assay was performed with Enhanced Cell Counting Kit-8 (Beyotime, C0043) according to manufacturer instruction. Briefly, 10 μl CCK-8 labelling reagent was added to each well and incubated at 37 °C with 5% $CO_2$ in a humidified atmosphere for 1 h. Cell viability was then determined using absorbance at 450 nm and evaluated at 0 h, 24 h and 48 h after transfection.

## DNA extraction

Genomic DNA was extracted using the MiniBEST Universal Genomic DNA Extraction kit (TaKaRa) and quantified using Qubit dsDNA HS Assay kit (Invitrogen) in a Qubit 3.0 fluorometer.

## GUIDE-seq

Genomic DNA was extracted at 48–72 h post transfection and 500 ng of DNA was used for next-generation sequencing library construction according to our previous work[26] with modification[65] (see updated dsODN sequences and amplification primers with discussion notes in Supplementary Table 10). Briefly, the library preparation procedure contains enzymatic fragmentation, end repair, A-tailing, adaptor ligation and two rounds of anchored nested PCR. The libraries were quantified with KAPA Library Quantification kits and sequenced on a NextSeq 1000 System (Illumina) using a 300-cycle kit (2× 150-bp paired-end) with standard Illumina sequencing workflow (that is, no need to adjust indexing cycles or use customized sequencing primers). Sequencing data (FASTQ files) were analysed using the GenomePAM pipeline with the off-target identification steps adapted from the GUIDE-seq pipeline (https://github.com/tsailabSJ/guideseq). The off targets were identified using the criteria of ≤6 mismatch bases with the intended targeting protospacer.

## GenomePAM Table

To identify enriched PAM motifs over genomic background, we developed the algorithm GenomePAM Table[66], involving the computational steps implemented in an R script to: (1) Identify the most significantly enriched single-base motif: we define the edited value as the sum of GUIDE-seq-detected genomic site numbers and GUIDE-seq read counts, with the latter linearly scaled to match the range of the former. The maximum value equals the highest number of genomic sites considered for all combinatorial potential motifs. Within the same motif window, a chi-square test is used to compare the edited value against the corresponding genomic background counts among all motifs. (2) Extend from the position identified in Step 1 bidirectionally: extend one base towards the 5′ end or one base towards the 3′ end and calculate the new edited values. Between the two extensions, the one with higher statistical significance is recorded and used for the next round of extension. (3) Repeat Step 2: continue extending in both directions until the ends of candidate bases are reached. Record all significant motifs without limiting motif length. (4) Report enriched motifs: report the enriched motifs along with the percentages of corresponding genomic sites edited, retaining only those motifs with increasing percentages from each iteration step (Fig. 1f).

## Reporting summary

Further information on research design is available in the Nature Portfolio Reporting Summary linked to this article.

## Data availability

Details of target sites identified have been included in the Supplementary Tables. Raw Fastq data are available at SRA (ID 1258724-BioProject-NCBI)[67]. Source data are provided with this paper.

## Code availability

The GenomePAM pipeline is available on GitHub[66]. The only input file required by the GenomePAM pipeline is the identifiedOfftargets.txt from the GUIDE-seq pipeline.

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

## Acknowledgements

We thank support from Lau Grant LC230003 (Z.Z.), Swedish Research Council 202001418 (Z.Z.); Research Grants Council of the Hong Kong Special Administrative Region 11103024 and T12-101/23-N (Z.Z.); Shenzhen Medical Research Fund B2402002 (Z.Z.); the InnoHK initiative of the Innovation and Technology Commission of the Hong Kong Special Administrative Region Government (Z.Z.); Lau Grant LC230002 (L.L.); the Kayden–Lambert MGH Research Scholar Award 2023–2028 (B.P.K.); and National Institutes of Health (NIH) grant DP2-CA281401 (B.P.K.). We thank O. W. C. Leung for proofreading the manuscript.

## Author contributions

M.Y. designed and performed the experiments and wrote the initial draft of the paper; L.A. contributed to the wet-lab work; B.W. contributed to experiment design, wet-lab work, data analysis and drafting of the paper; S.L., L.I. and J.L. performed bioinformatic and statistical analyses; L.L. contributed to funding acquisition and project supervision; S.Q.T. contributed to project supervision; B.P.K. contributed to funding acquisition and project supervision; and Z.Z. conceived of the method, obtained funding and supervised the project; all authors reviewed and confirmed the submitted version of the paper.

## Funding

## Competing interests

Z.Z. is a co-founder and a scientific advisor for and holds equity in GenEditBio whose interest is reviewed and regulated by institutional Outside Practice policies annually. Z.Z. and M.Y. are inventors on a patent (WIPO Patent application no. WO2024230784A1, 2024) for GenomePAM technologies. B.P.K. is an inventor on patents or patent applications (WIPO Patent application no. WO2021151065A2, 2021) filed by MGB for HT-PAMDA technologies that describe genome engineering technologies and methods to characterize the PAM. B.P.K. consults for Novartis Venture Fund, Foresite Labs and Jumble Therapeutics, and is on the scientific advisory board of Acrigen Biosciences, Life Edit Therapeutics and Prime Medicine. B.P.K. has a financial interest in Prime Medicine, Inc., a company developing therapeutic CRISPR-Cas technologies for gene editing. B.P.K.'s interests were reviewed and are managed by MGH and MGB in accordance with their conflict-of-interest policies. S.Q.T. is a co-inventor on a patent (WIPO Patent application no. WO2015200378A1, 2015) for GUIDE-seq, and a member of the scientific advisory boards of Ensoma and Prime Medicine. B.W., L.I. and J.L. are employees of GenEditBio. The other authors declare no competing interests.

## Additional information

**Extended data** is available for this paper at https://doi.org/10.1038/s41551-025-01464-y.

**Correspondence and requests for materials** should be addressed to Zongli Zheng.

[1]Department of Biomedical Sciences and Tung Biomedical Sciences Centre, College of Biomedicine, City University of Hong Kong, Kowloon, Hong Kong SAR, China. [2]Ming Wai Lau Centre for Reparative Medicine, Karolinska Institutet, Shatin, Hong Kong SAR, China. [3]GenEditBio Limited, Shatin, Hong Kong SAR, China. [4]Center for Neuromusculoskeletal Restorative Medicine, Hong Kong Science Park, Shatin, Hong Kong SAR, China. [5]Department of Neuroscience, Karolinska Institutet, Stockholm, Sweden. [6]Department of Surgery, Prince of Wales Hospital, The Chinese University of Hong Kong, Shatin, Hong Kong SAR, China. [7]Department of Hematology, St. Jude Children's Research Hospital, Memphis, TN, USA. [8]Center for Genomic Medicine, Massachusetts General Hospital, Boston, MA, USA. [9]Department of Pathology, Massachusetts General Hospital, Boston, MA, USA. [10]Department of Pathology, Harvard Medical School, Boston, MA, USA. [11]Department of Precision Diagnostic and Therapeutic Technology Biotechnology and Health Centre, City University of Hong Kong Shenzhen Research Institute, Shenzhen, China. [12]Department of Medical Epidemiology and Biostatistics, Karolinska Institutet, Stockholm, Sweden. ✉e-mail: Zongli.Zheng@cityu.edu.hk

| Name | Sequence (5' - 3') | Occurrences in human genome (hg38) | Flanking sequence diversity | Note |
|---|---|---|---|---|
| Rep-1 | GTGAGCCACTGTGCCTGGCC | 8,471 | | For 3' PAM |
| Rep-1RC | GGCCAGGCACAGTGGCTCAC | 8,471 | | For 5' PAM |
| Rep-2 | GAGCCACCATGCCTGGCCAA | 1,991 | | Not ideal, uneven bases at positions 9 and 10 |
| Rep-2RC | TTGGCCAGGCATGGTGGCTC | 1,991 | | Not ideal, uneven bases at positions -9 and -10 |
| Rep-3 | GTGAGCCACCGCACCTGGCC | 5,856 | | For 3' PAM |
| Rep-3RC | GGCCAGGTGCGGTGGCTCAC | 5,856 | | For 5' PAM |
| Rep-4 | GTGAGCCACCGCGCCTGGCC | 9,089 | | For 3' PAM |
| Rep-4RC | GGCCAGGCGCGGTGGCTCAC | 9,089 | | For 5' PAM |
| Rep-U1 | CACACACACACACACACACA | 505,405 | | Unsuitable, too simple sequence |
| Rep-U2 | CTCCCAAAGTGCTGGGATTA | 356,589 | | Unsuitable, low-diversity flanking sequence |

**Extended Data Fig. 1 | Occurrence and flanking sequence diversity of selected repetitive sequences.** The list shows 10 example repetitive sequences (Rep-1, Rep-1RC, Rep-2, Rep-2RC, Rep-3, Rep-3RC, Rep-4, Rep-4RC, Rep-U1 and Rep-U2), with their occurrences in human genome (hg38), diversity of flanking 10 bases and explaining notes for their suitabilities to be used as spacer for GenomePAM analysis.

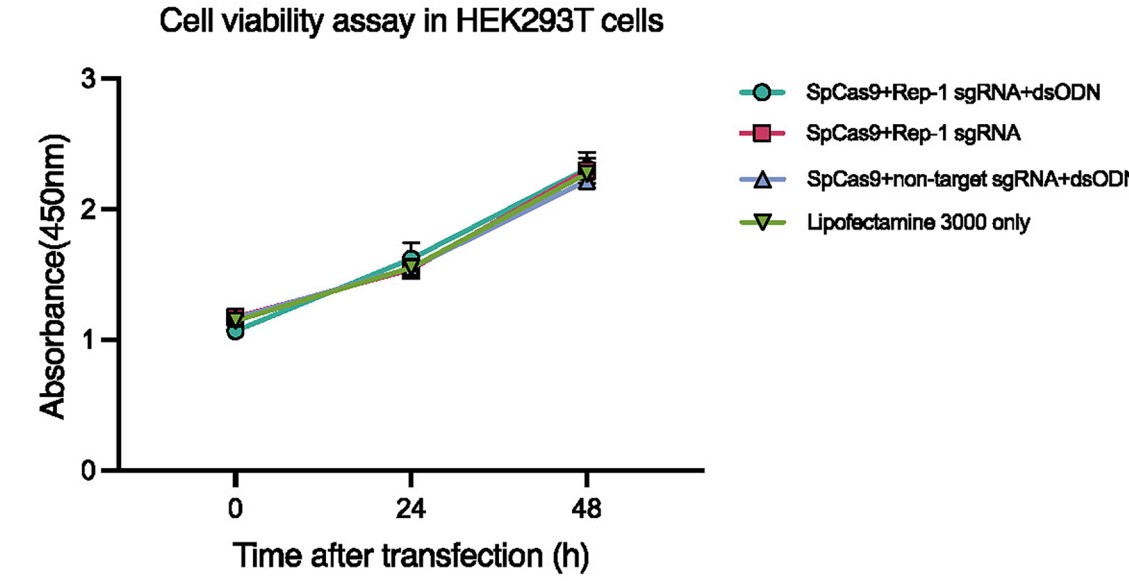

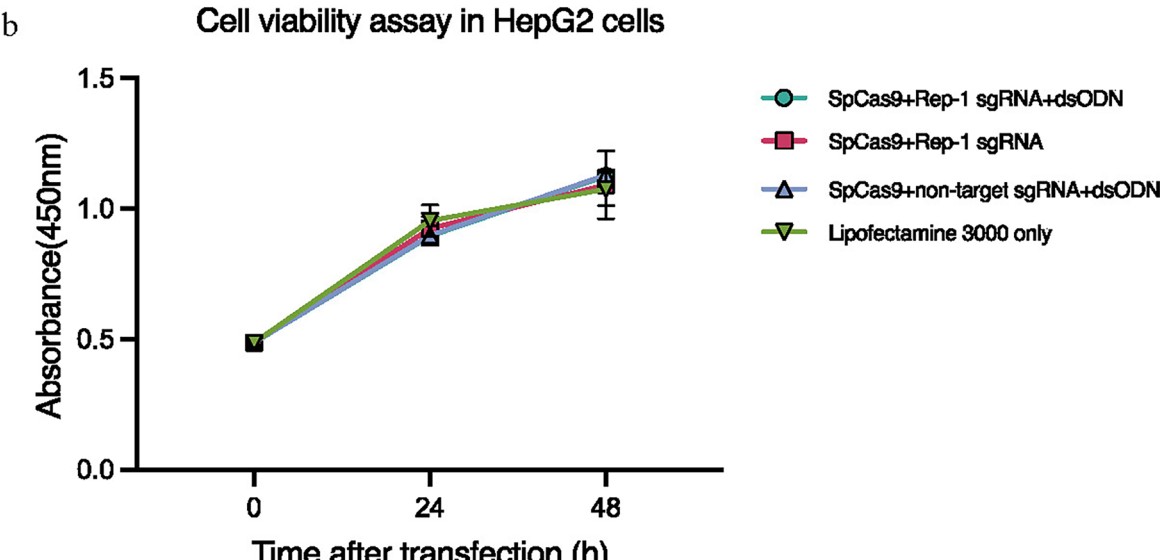

**Extended Data Fig. 2 | Cell viability assay when transfected with four different conditions in HEK293T (a) and HepG2 (b) cells.** 1) SpCas9 plasmid + Rep-1 sgRNA plasmid + dsODN; 2) SpCas9 plasmid + Rep-1 sgRNA plasmid; 3) SpCas9 plasmid + non-target sgRNA plasmid + dsODN; and 4) Lipofectamine 3000 only. Absorbance at 450 nm were evaluated 0 h, 24 h and 48 h after transfection.

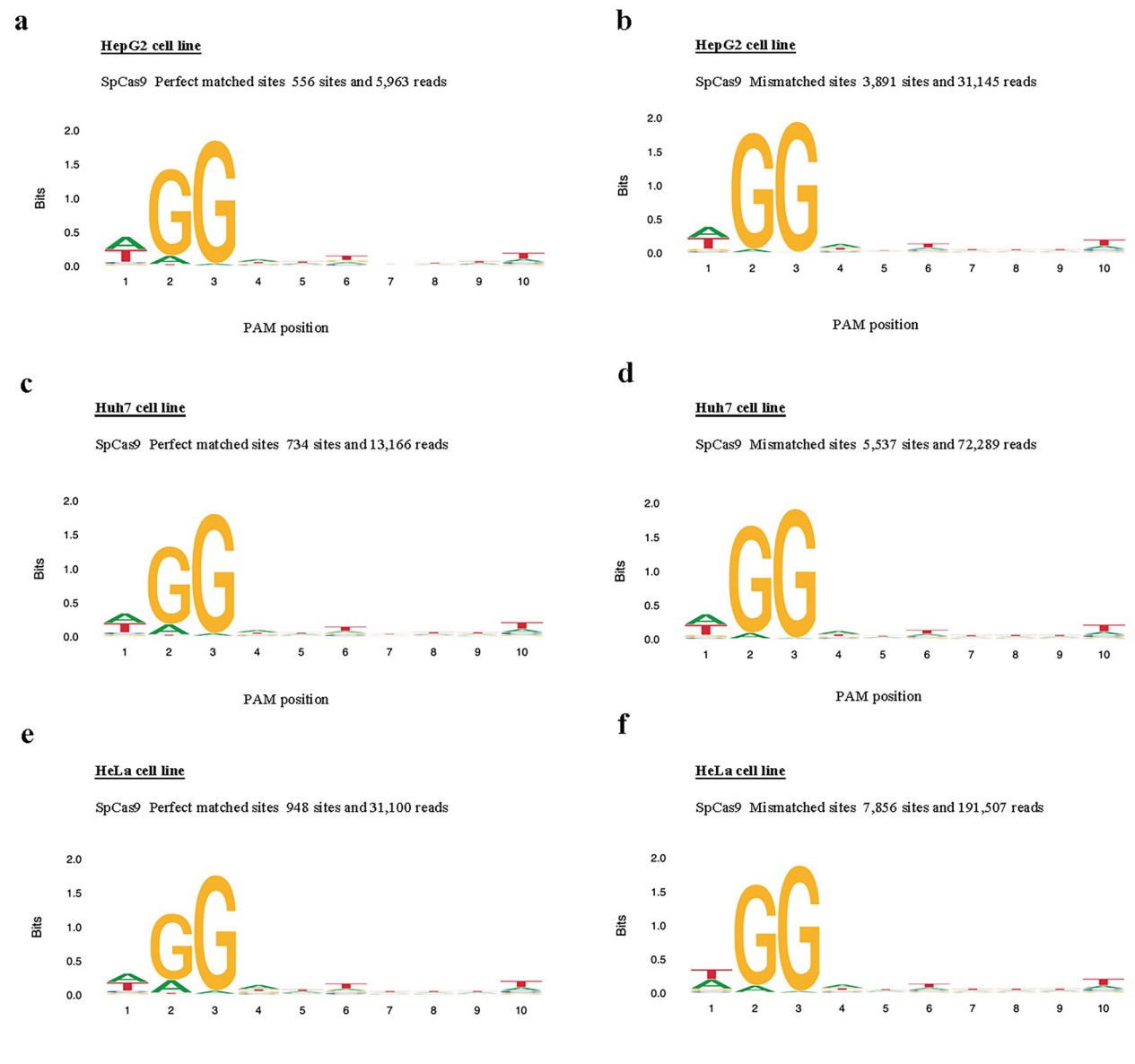

**Extended Data Fig. 3 | Evaluation of GenomePAM on SpCas9 in three other cell lines than HEK293T.** SeqLogo results for SpCas9 proteins PAM preferences in HepG2 (**a, b**), Huh7 (**c, d**) and HeLa (**e, f**) cells. (**a, c, e**) were summarized by associated perfect match spacers and (**b, d, f**) mismatch spacers.

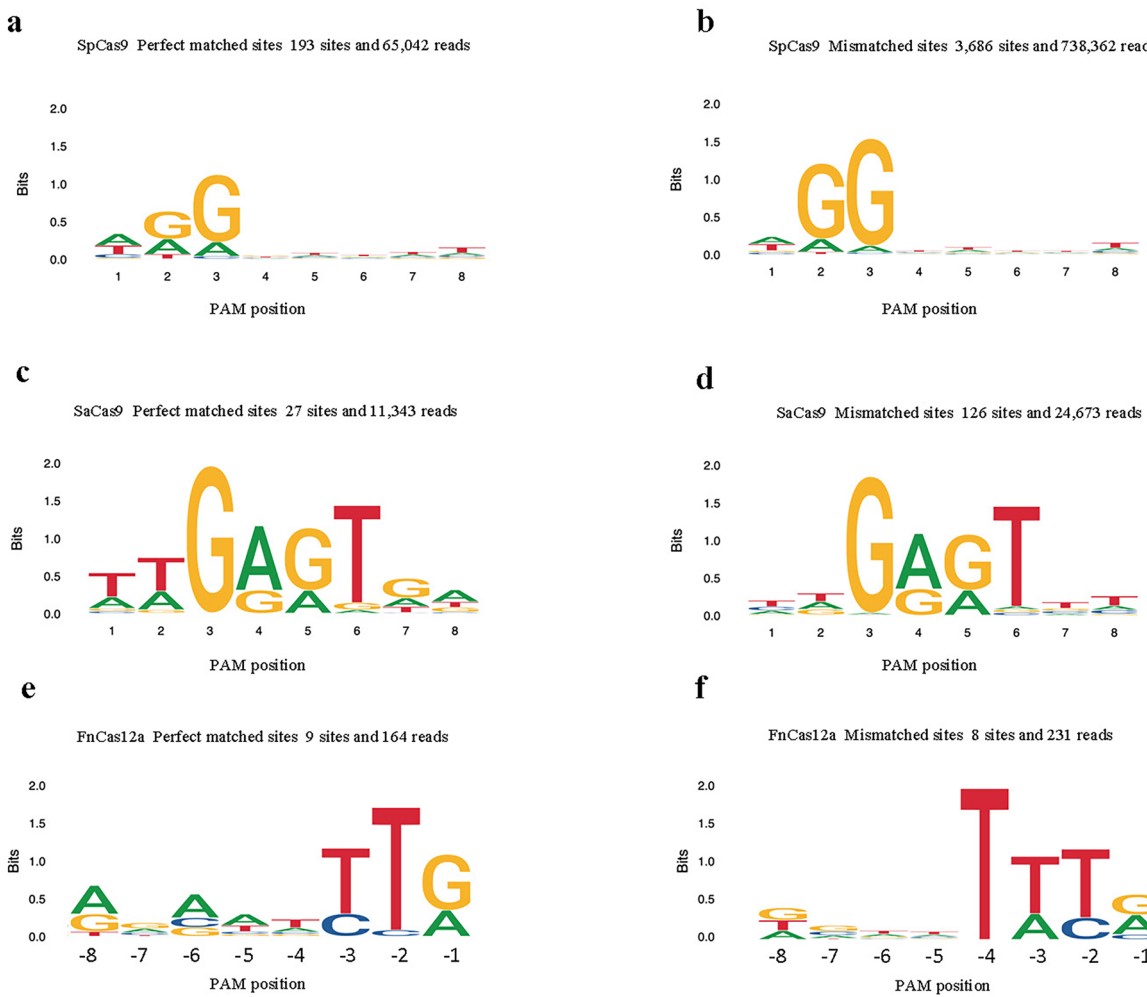

**Extended Data Fig. 4 | GenomePAM results of SpCas9 and SaCas9 using Rep-2 as the spacer, and FnCas12a using Rep-2RC as the spacer in HEK293T cells.** See Extended Data Fig. 1 for the sequences of Rep-2 and Rep-2RC. The PAMs were summarized by associated perfect match spacers (**a, c, e**) and by mismatch spacers (**b, d, f**).

TiCas9 K1315A Perfect match: 121 sites and 36,276 reads

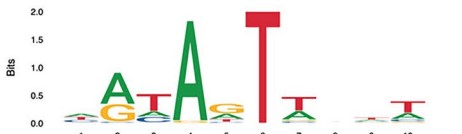

TiCas9 K1315C Perfect match: 97 sites and 21,056 reads

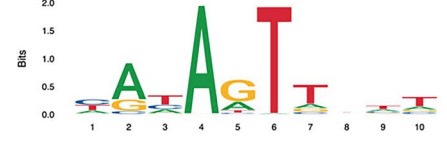

TiCas9 K1315D Perfect match: 14 sites and 3,397 reads

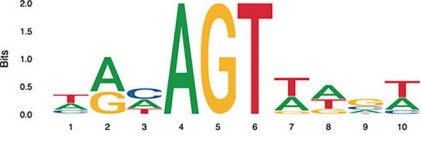

TiCas9 K1315E Perfect match: 25 sites and 4,152 reads

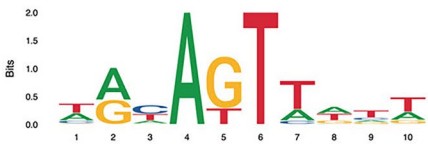

TiCas9 K1315F Perfect match: 17 sites and 5,690 reads

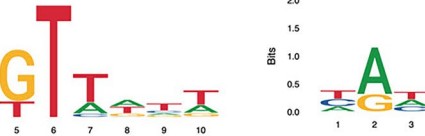

TiCas9 K1315G Perfect match: 58 sites and 16,933 reads

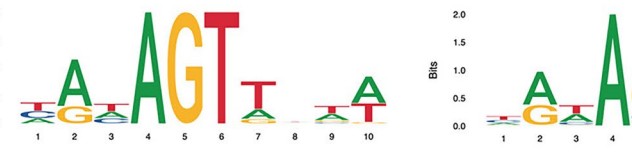

TiCas9 K1315H Perfect match: 27 sites and 8,324 reads

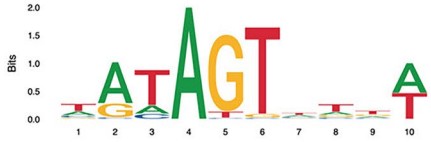

TiCas9 K1315I Perfect match: 1 site and 2 reads

TiCas9 K1315L Perfect match: 89 sites and 26,968 reads

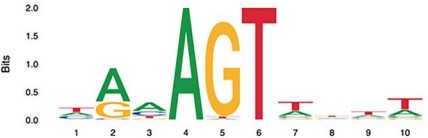

TiCas9 K1315M Perfect match: 4 sites and 106 reads

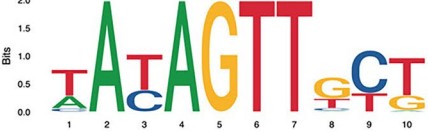

TiCas9 K1315N Perfect match: 174 sites and 9,321 reads

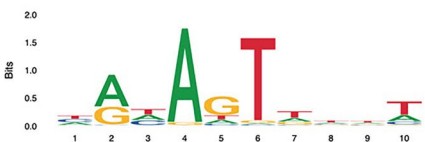

TiCas9 K1315P Perfect match: 7 sites and 56 reads

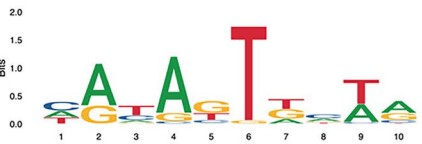

TiCas9 K1315R Perfect match: 28 sites and 8,680 reads

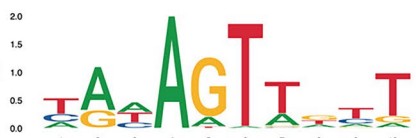

TiCas9 K1315S Perfect match: 33 sites and 739 reads

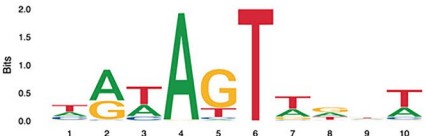

TiCas9 K1315T Perfect match: 109 sites and 3,584 reads

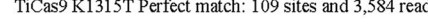
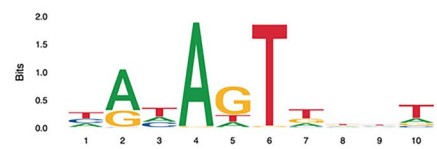

TiCas9 K1315V Perfect match: 446 sites and 18,948 reads

TiCas9 K1315W Perfect match: 60 sites and 2,711 reads

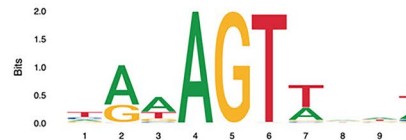

TiCas9 K1315Y Perfect match: 20 sites and 7,157 reads

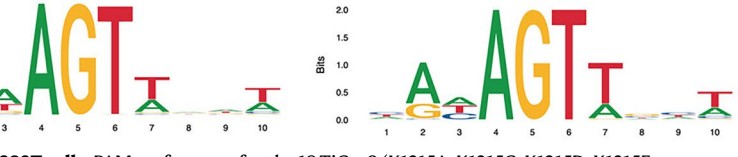

**Extended Data Fig. 5 | GenomePAM analyses on 18 TiCas9 variants in HEK293T cells.** PAM preferences for the 18 TiCas9 (K1315A, K1315C, K1315D, K1315E, K1315F, K1315G, K1315H, K1315I, K1315L, K1315M, K1315N, K1315P, K1315R, K1315S, K1315T, K1315V, K1315W and K1315Y) were summarized by associated perfect matched spacers.

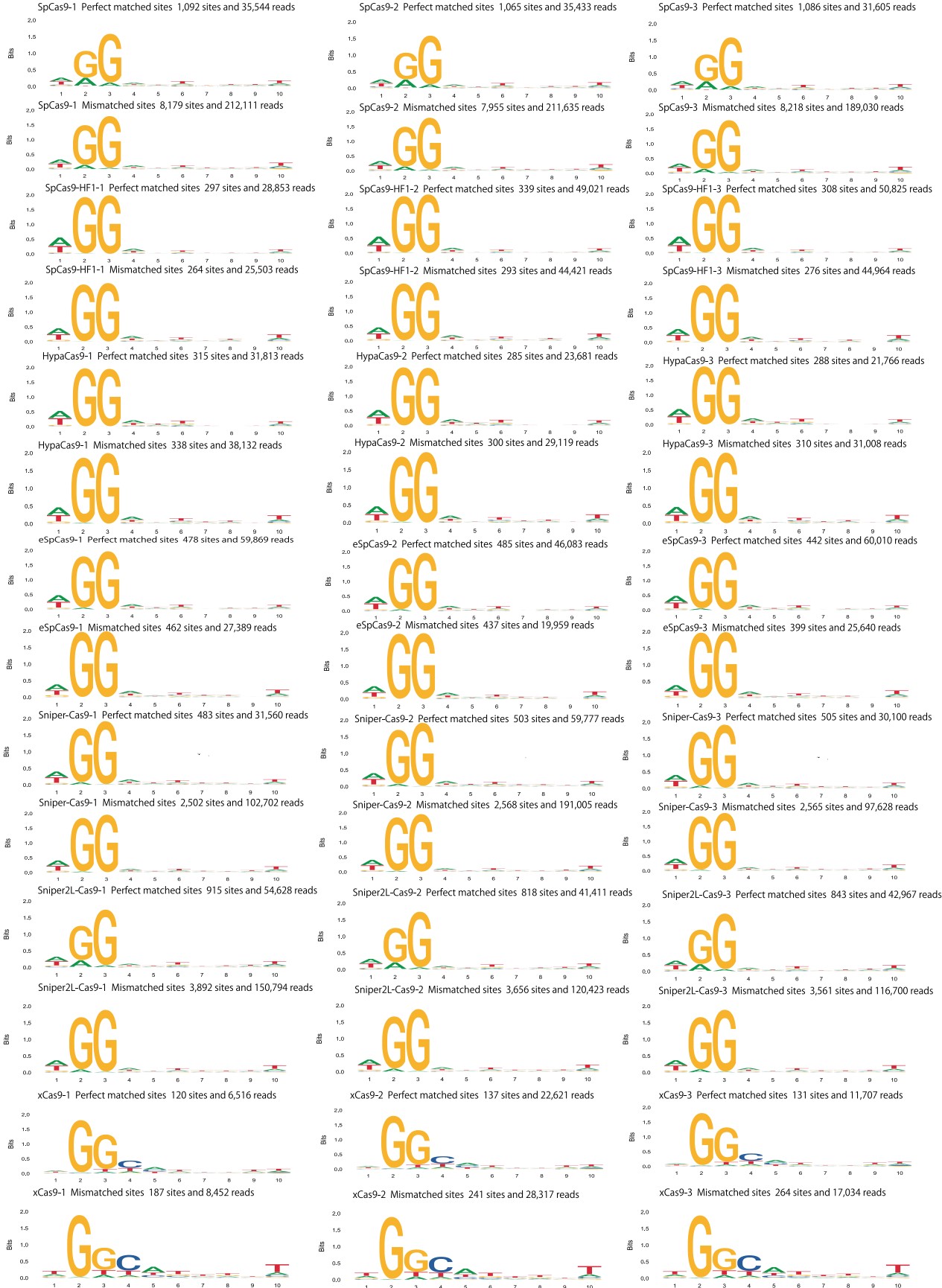

**Extended Data Fig. 6 | Using Rep-1 as the spacer, evaluation of Genome-PAM on seven variants of SpCas9 (SpCas9, SpCas9-HF1, HypaCas9, eSpCas9(1.1), Sniper-Cas9, Sniper2L-Cas9, and xCas9) in HEK293T cells.** PAM preferences were summarized by associated perfect match spacers and mismatch spacers, respectively.

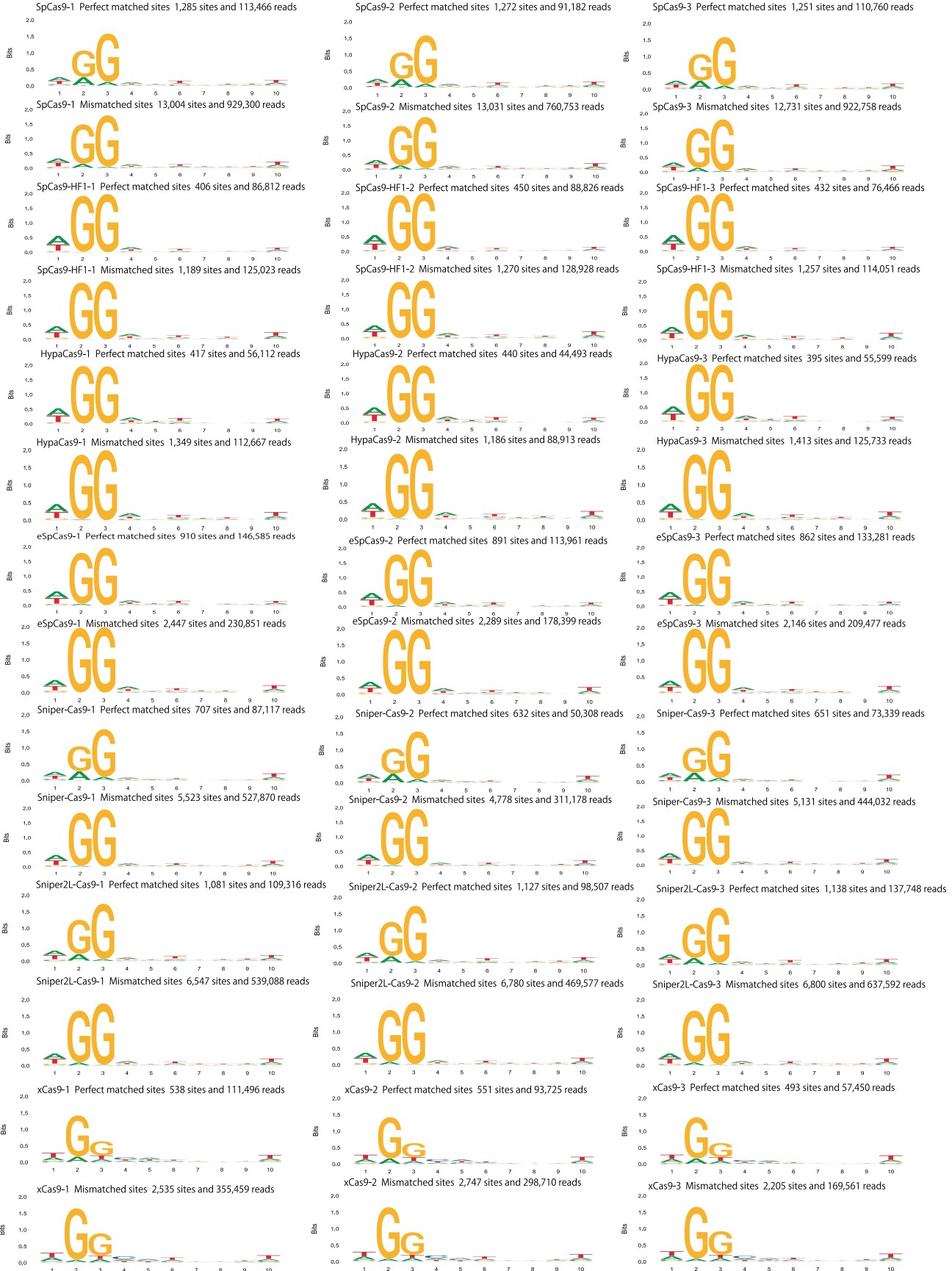

**Extended Data Fig. 7 | Using Rep-3 as the spacer, evaluation of Genome-PAM on seven variants of SpCas9 (SpCas9, SpCas9-HF1, HypaCas9, eSpCas9(1.1), Sniper-Cas9, Sniper2L-Cas9, and xCas9) in HEK293T cells.** PAM preferences were summarized by associated perfect match spacers and mismatch spacers, respectively.

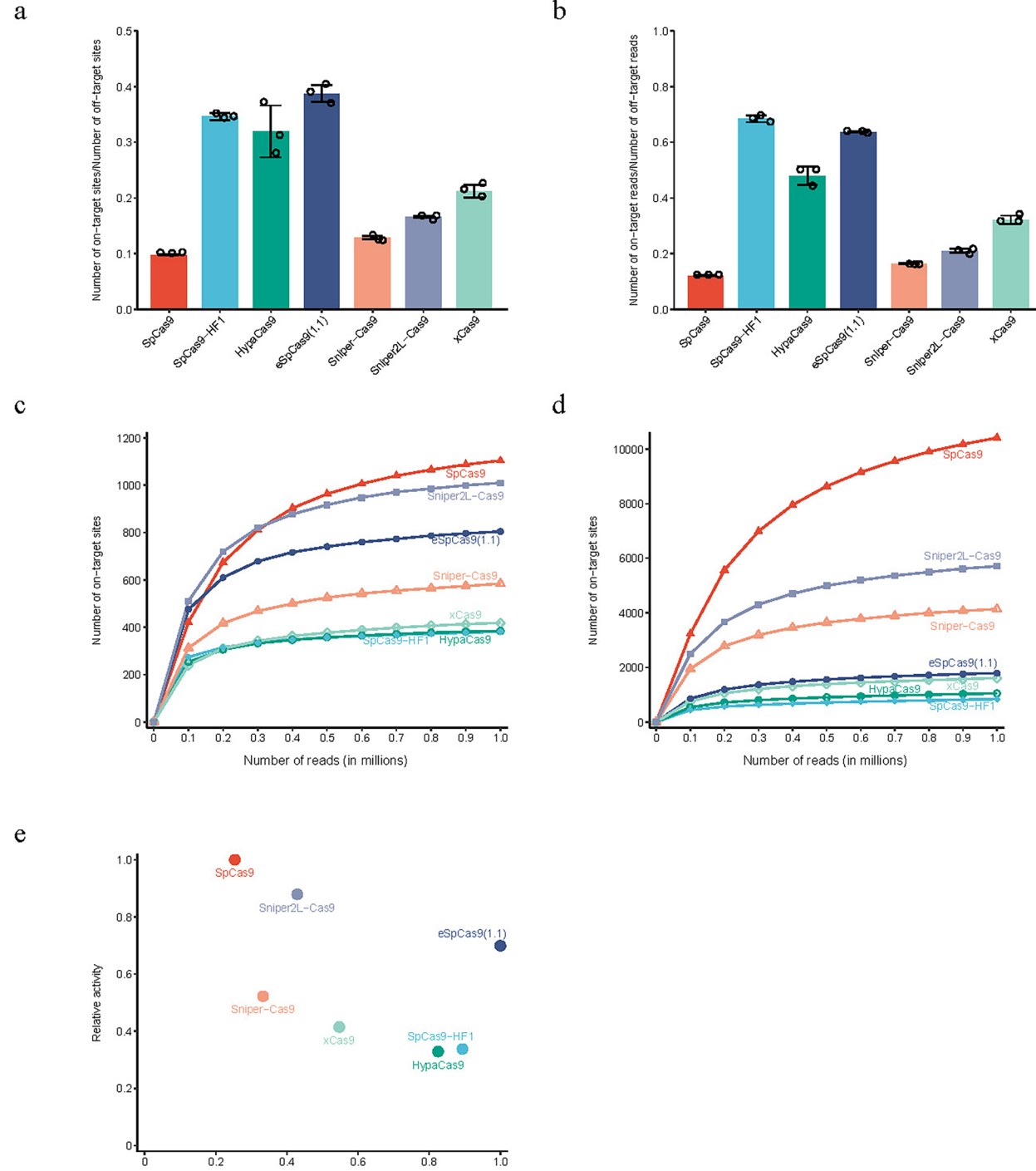

**Extended Data Fig. 8 | General activities and specificities of different SpCas9 variants using Rep-3 as the spacer in GenomePAM assay. (a)** The number of on-target sites to the number of off-target sites of seven SpCas9 variants. **(b)** The number of on-target sequencing reads to the number of off-target sequencing reads of seven SpCas9 variants. The number of on-target sites **(c)** and the number of off-target sites **(d)** detected when using randomly down-sampled datasets, from 0.1 M to 1 M raw sequencing reads. **(e)** Relative activities (defined as the number of perfect match sites relative to that of the WT SpCas9) and specificities (defined as the ratio of perfect match to mismatch site numbers relative to the ratio in SpCas9-HF1) of seven SpCas9 variants.

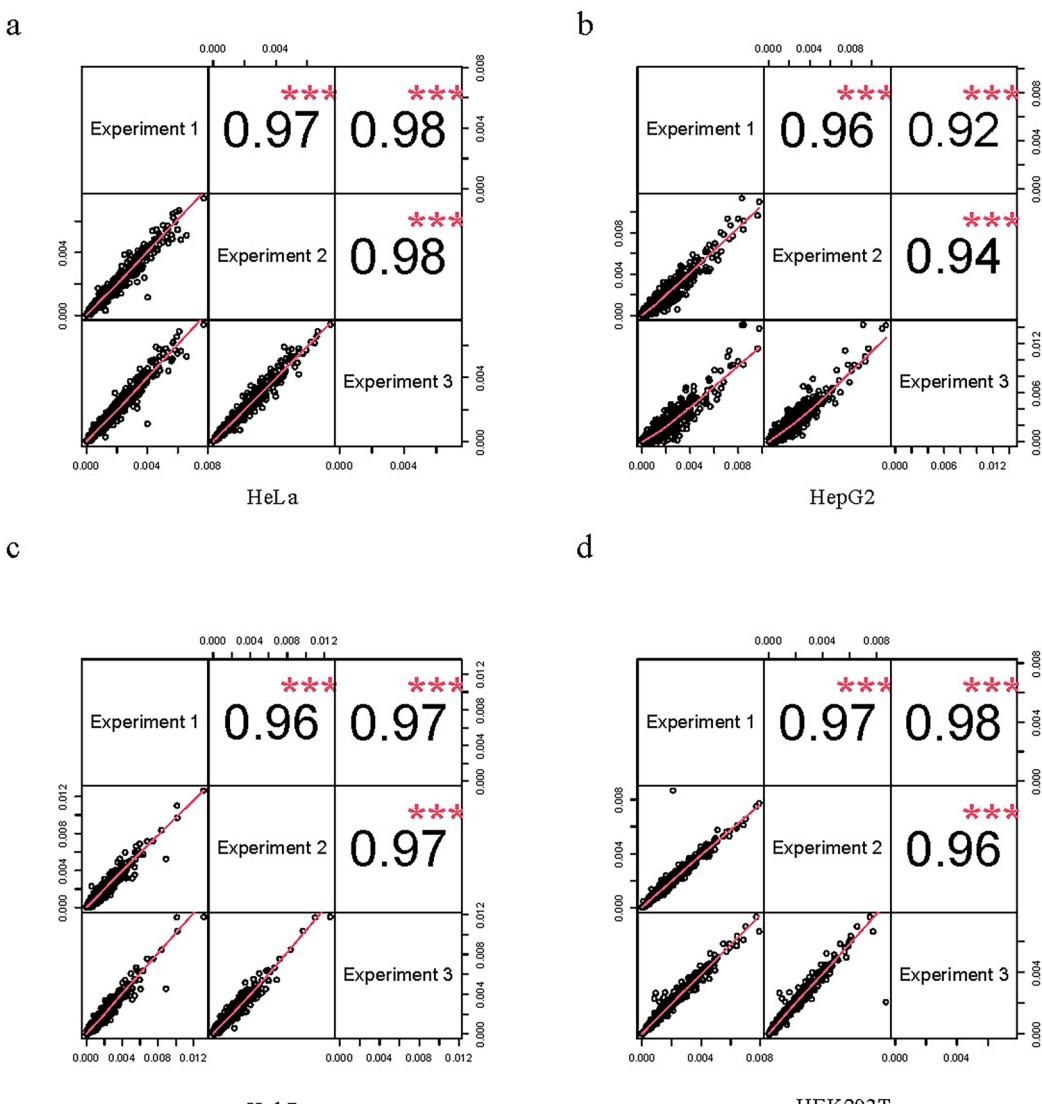

**Extended Data Fig. 9 | The correlations among three replicates of genome-wide chromatin accessibility for HeLa cells (a), HepG2 cells (b), Huh7 cells (c) and HEK293T cells (d).** The number represented the Pearson correlation coefficient. Each point represents the proportion of reads in each bin.

# Reporting Summary

## Statistics

For all statistical analyses, confirm that the following items are present in the figure legend, table legend, main text, or Methods section.

| n/a | Confirmed | |
|---|---|---|
| ☐ | ☒ | The exact sample size (*n*) for each experimental group/condition, given as a discrete number and unit of measurement |
| ☐ | ☒ | A statement on whether measurements were taken from distinct samples or whether the same sample was measured repeatedly |
| ☐ | ☒ | The statistical test(s) used AND whether they are one- or two-sided<br>*Only common tests should be described solely by name; describe more complex techniques in the Methods section.* |
| ☒ | ☐ | A description of all covariates tested |
| ☒ | ☐ | A description of any assumptions or corrections, such as tests of normality and adjustment for multiple comparisons |
| ☐ | ☒ | A full description of the statistical parameters including central tendency (e.g. means) or other basic estimates (e.g. regression coefficient) AND variation (e.g. standard deviation) or associated estimates of uncertainty (e.g. confidence intervals) |
| ☐ | ☒ | For null hypothesis testing, the test statistic (e.g. *F*, *t*, *r*) with confidence intervals, effect sizes, degrees of freedom and *P* value noted<br>*Give P values as exact values whenever suitable.* |
| ☒ | ☐ | For Bayesian analysis, information on the choice of priors and Markov chain Monte Carlo settings |
| ☒ | ☐ | For hierarchical and complex designs, identification of the appropriate level for tests and full reporting of outcomes |
| ☒ | ☐ | Estimates of effect sizes (e.g. Cohen's *d*, Pearson's *r*), indicating how they were calculated |

*Our web collection on statistics for biologists contains articles on many of the points above.*

## Software and code

Policy information about availability of computer code

| | |
|---|---|
| Data collection | The study developed new codes: https://github.com/Zheng-NGS-Lab/genomePAM; All datasets generated by this study have been deposited at SRA (ID 1258724 - BioProject - NCBI) (https://www.ncbi.nlm.nih.gov/bioproject/PRJNA1258724). Human genome reference hg38 was collected from UCSC (https://hgdownload.soe.ucsc.edu/goldenpath/hg38/bigZips/). |
| Data analysis | https://github.com/tsailabSJ/guideseq; BWA version: 0.7.17-r1188; Samtools version: 1.8 (htslib 1.8). R version 4.1.2; ggseqlogo version 0.2. |

For manuscripts utilizing custom algorithms or software that are central to the research but not yet described in published literature, software must be made available to editors and reviewers. We strongly encourage code deposition in a community repository (e.g. GitHub). See the Nature Portfolio guidelines for submitting code & software for further information.

## Data

Policy information about availability of data

All manuscripts must include a data availability statement. This statement should provide the following information, where applicable:
- Accession codes, unique identifiers, or web links for publicly available datasets
- A description of any restrictions on data availability
- For clinical datasets or third party data, please ensure that the statement adheres to our policy

Detail target sites identified have been included in the supplementary Excel files. Raw Fastq data will be deposit at SRA before publication.

# Research involving human participants, their data, or biological material

Policy information about studies with <u>human participants or human data</u>. See also policy information about <u>sex, gender (identity/presentation), and sexual orientation</u> and <u>race, ethnicity and racism</u>.

| Reporting on sex and gender | N/A |
|---|---|
| Reporting on race, ethnicity, or other socially relevant groupings | N/A |
| Population characteristics | N/A |
| Recruitment | N/A |
| Ethics oversight | N/A |

Note that full information on the approval of the study protocol must also be provided in the manuscript.

# Field-specific reporting

Please select the one below that is the best fit for your research. If you are not sure, read the appropriate sections before making your selection.

☒ Life sciences  ☐ Behavioural & social sciences  ☐ Ecological, evolutionary & environmental sciences

For a reference copy of the document with all sections, see <u>nature.com/documents/nr-reporting-summary-flat.pdf</u>

# Life sciences study design

All studies must disclose on these points even when the disclosure is negative.

| Sample size | The is one place in our manuscript mentioning 'n', in Line 281: "large-scale synthetic oligos (n = 26,891)". This was from a previous study that we cited, to demonstrate the advantage of our study without the need to synthesis any oligo but to use sequence already in the genome. |
|---|---|
| Data exclusions | No data were excluded from the analyses |
| Replication | All experiments were replicated at least 3. |
| Randomization | The study did not involve human or animal subjects. All cell line experiments started with relevant cell lines and were replicated at least 3 times. |
| Blinding | The study did not involve human or animal subjects. All cell line experiments started with relevant cell lines and were replicated at least 3 times. |

# Reporting for specific materials, systems and methods

We require information from authors about some types of materials, experimental systems and methods used in many studies. Here, indicate whether each material, system or method listed is relevant to your study. If you are not sure if a list item applies to your research, read the appropriate section before selecting a response.

## Materials & experimental systems

| n/a | Involved in the study |
|---|---|
| ☒ | ☐ Antibodies |
| ☐ | ☒ Eukaryotic cell lines |
| ☒ | ☐ Palaeontology and archaeology |
| ☒ | ☐ Animals and other organisms |
| ☒ | ☐ Clinical data |
| ☒ | ☐ Dual use research of concern |
| ☒ | ☐ Plants |

## Methods

| n/a | Involved in the study |
|---|---|
| ☒ | ☐ ChIP-seq |
| ☒ | ☐ Flow cytometry |
| ☒ | ☐ MRI-based neuroimaging |

## Eukaryotic cell lines

Policy information about cell lines and Sex and Gender in Research

| | |
|---|---|
| Cell line source(s) | All cell lines were purchased from ATCC (American type culture collection). HEK293T, HepG2, Huh7 and HeLa |
| Authentication | All cell lines used were authenticated by STR (Short Tandem Repeat) when purchased from ATCC |
| Mycoplasma contamination | All cell lines were tested negative for mycoplasma contamination |
| Commonly misidentified lines (See ICLAC register) | No commonly misidentified cell lines were used in the study |

## Plants

| | |
|---|---|
| Seed stocks | N/A |
| Novel plant genotypes | N/A |
| Authentication | N/A |

