## [Peer Review File · Nature Biomedical Engineering]

GenomePAM directs PAM characterization and engineering of CRISPR-Cas nucleases using mammalian genome repeats

Corresponding Author: Dr Zongli Zheng

Version 0:

Decision Letter:

Dear Zongli,

Thank you again for submitting to *Nature Biomedical Engineering* your manuscript, "Mammalian genome repeats for direct PAM requirement characterization and activity profiling of CRISPR-Cas nucleases". The manuscript has been seen by three experts, whose reports you will find at the end of this message.

You will see that the reviewers appreciate the work. However, they express concerns about the advantages utility of the method, and in this regard provide useful suggestions for improvement. We hope that with substantial further work you can address the criticisms and convince the reviewers of the merits of the study. In particular, we would expect that a revised version of the manuscript provides:

- * Evidence of advantageous utility of the method, particularly by benchmarking it against competing techniques, such as PAM-DOSE.
- * Extended evidence of the method's robustness and discussion of its current limitations, as per the various relevant questions from the reviewers.
- * Thorough methodological reporting.

When you are ready to resubmit your manuscript, please upload the revised files, a point-by-point rebuttal to the comments from all reviewers, the [reporting summary](https://www.nature.com/authors/policies/ReportingSummary.pdf), and a cover letter that explains the main improvements included in the revision and responds to any points highlighted in this decision.

Please follow the following recommendations:

- * Clearly highlight any amendments to the text and figures to help the reviewers and editors find and understand the changes (yet keep in mind that excessive marking can hinder readability).
- * If you and your co-authors disagree with a criticism, provide the arguments to the reviewer (optionally, indicate the relevant points in the cover letter).
- * If a criticism or suggestion is not addressed, please indicate so in the rebuttal to the reviewer comments and explain the reason(s).
- * Consider including responses to any criticisms raised by more than one reviewer at the beginning of the rebuttal, in a section addressed to all reviewers.
- * The rebuttal should include the reviewer comments in point-by-point format (please note that we provide all reviewers will the reports as they appear at the end of this message).
- * Provide the rebuttal to the reviewer comments and the cover letter as separate files.

We expect that you will be able to resubmit the manuscript within **12 weeks** of receiving this message. If this is the case, you will be protected against potential scooping. Otherwise, we will be happy to consider a revised manuscript as long as the significance of the work is not compromised by work published elsewhere or accepted for publication at *Nature Biomedical Engineering*.

We hope that you will find the referee reports helpful when revising the work. Please do not hesitate to contact me should you have any questions.

Best wishes,

Pep

Pep Pàmies

Chief Editor, Nature Biomedical Engineering

Reviewer #1 (Report for the authors (Required)):

PAM is crucial for CRISPR tools to locate target sites. Various CRISPR tools have evolved to utilize different PAMs, making the identification of accurate PAMs for new CRISPR tools essential. Several strategies exist for identifying PAMs, and in this paper, Yu et al. propose a new and smart strategy named genomePAM, which utilizes endogenous genomic repeat sequences. This method is advantageous as it reduces the need to design diverse PAM-associated sequences. However, several important concerns must be addressed before it can become a very useful tool:

Main concerns:

1. The genomePAM method integrates GUIDE-seq, which requires skills to handle and is more complex than the target sequencing methods widely used in other PAM-identification techniques. The authors should discuss this point in the Discussion section. Moreover, while AMP-seq is mentioned as a potential alternative to GUIDE-seq, no AMP-seq data is provided. The authors should test and present the AMP-seq-involved version.
2. Several factors make data normalization complicated and may affect the accuracy of identified PAMs. The authors should address these challenges with new data or discuss them in more details.
 - 2.1 As shown in Figure 6, different target sites exhibit varying levels of chromatin accessibility. Therefore, each site should have a different weight when calculating the consensus sequence of the PAM. Did the authors use some way to normalize the accessibility?
 - 2.2 Editing Alu repeats may affect cell activity, potentially impacting cell proliferation or viability. This is a concern for all endogenous sites. The authors should consider whether there are sites in non-functional regions or elements that could mitigate this issue.
 - 2.3 From Figure 1C, it appears that the spacer sequences of used repeats contain diverse mismatches, which may affect the cleavage frequency of different sites. The authors need to explain how they normalize this variability.
 - 2.4 Whether it's necessary to use different repeat spacer sequences to verify the identified PAMs to exclude the impact of single spacer sequence?
3. When comparing the fidelity of several high-fidelity Cas9 variants, the variation in PAM and spacer sequences across different sites makes the comparison of cleavage activity at off-target sites complicated. The authors should analyze and show the diversity of combined spacer sequences and PAMs to ensure sufficient diversity for this experiment.

Minor points:

1. The names of several methods are abbreviated in the Introduction (HT-PAMDA, GUIDE-seq, AMP-seq). The full names should be provided at their first occurrence.
2. Schematics are needed for all CRISPR tools to explain the position of PAMs relative to the spacer sequence. This is particularly important because the position of PAMs varies, as seen with CjCas9.

Reviewer #2 (Report for the authors (Required)):

In this work, Yu and coworkers devise a novel approach to characterize CRISPR PAMs and off-targets using repetitive sequences in the human genome. Existing approaches for PAM determination involve large libraries of randomized sequences and are normally performed outside of human cells. The authors' approach, termed GenomePAM, targets known repetitive sequences with highly variable flanking ends. After targeting this site with a Cas nuclease, the flanking sequences that allowed DNA cleavage are determined based on existing approaches such as GUIDE-seq or AMP-seq that integrate a DNA barcode at a cut site and sequence the surrounding region. The authors validate GenomePAM using six established Cas nucleases, and they apply the approach to a previously uncharacterized Cas12a nuclease. The authors next take advantage of the repetitive elements containing some sequence variations, offering a simpler screen to gauge the propensity

for off-targeting. Finally, by comparing cleaved sites across cell lines, the authors identify cell line-specific sites presumably affected by the chromatin state.

Within the large number of PAM determination approaches that already exist, GenomePAM is a unique and inventive entry differentiated by its ability to rely on existing sequence conservation in human cells and its ability to determine PAMs and off-target propensities in the very environment in which the nucleases would be applied. At the same time, there are notable limitations to the approach, and the general work lacks depth to argue for GenomePAM over existing methodologies. The application of GenomePAM to a new Cas nuclease was helpful, although the nuclease in itself was not a notable addition to the massive existing set of Cas nucleases.

Major comments:

1. GenomePAM does come with important limitations that should be addressed by the authors. Specifically, the PAM library is much smaller than what would be used in all other assays, this library is further reduced by inaccessible sites (e.g., in heterochromatin), and it requires Guide-Seq or AMP-seq that are much more complicated to perform and analyze than traditional amplicon-based sequencing used with all other methodologies.

2. An initially attractive feature of the methodology is that it's entirely performed in human cells, allowing the authors to gain context-specific insights free of potential biases in other contexts (e.g., in vitro assays). However, the extracted insights at best match what was extracted from non-mammalian approaches, suggesting that prior approaches would be sufficient. Providing an example where the mammalian insights make a difference would be incredibly helpful to support the methodology; otherwise, I expected the field continue using much easier approaches that do not rely on complicated sequencing approaches like GUIDE-Seq and AMP-seq.

3. The authors also argue that traditional randomized libraries have strong biases or are difficult to maintain. However, this issue is a relatively minor one, particularly for PAM screening. In addition, the set of available sequences flanking the repetitive elements likely also has strong biases on top of covering only a small percentage of possible sequences. Here, a direct comparison to a complete PAM library would be helpful.

4. For all off-target analyses, I had difficulty understanding what is defined as an off-target and whether additional insights can be drawn beyond a given nuclease being more or less potent or specific. For instance, is it clear where mismatches are accommodated across the guide? And how would these biases change when starting with an entirely distinct target (e.g., Rep-2)? Finally, does the screen provide at least equivalent information compared to existing off-target screens that can inform off-target prediction?

5. In general, analyses throughout the work could use much more depth. For instance, how many different repetitive sequences were identified, what is their frequency, distribution, and associated PAM profile? What biases or unexpected cleaved sequences were identified for each nuclease, and what fraction of sites that should have been cleaved but weren't (e.g., due to chromatin accessibility)? Finally, how do the accessible and inaccessible sites align with heterochromatin in these cell lines (since chromatin is never directly mapped)?

Other comments:

6. In describing different existing PAM determination methods, one particularly absent set involve cell-free transcription-translation. These approaches overcome many of the hurdles noted for in vitro and in vivo methods, even if they are not in a mammalian context. At the very least, two important reports of these methods should be incorporated: PMID = 29304331, 35216669.

7. What other repetitive sequences were identified? The authors briefly mention Rep-2, although a more systematic presentation is needed. These analyses would be good to include in the SI.

8. In some places the authors use the term Cpf1, which is now outdated and no longer used. Cas12a should be used instead in line with the unified naming of Cas nucleases.

9. Use prime symbols rather than apostrophes when referring to either end of a nucleic acid.

10. L. 150: In this section, SpCas9 was included in the validation set, yet it was already demonstrated in the prior section. Some rewording is in order.

11. L. 168: FnCas12a is known to be biased against a T at the -1 position. To what extent was this observed? As this bias is common across Cas12a nucleases, is it also seen with RuCas12a?

12. L. 187-188: does this refer to repetitive sequences with some deviations? Some clarification is necessary.

13. L. 194-195: it's worth acknowledging that many other PAM determination methodologies involve positive screens and thus isn't a unique feature of GenomePAM.

14. L. 200-202: some rewording needed, as GenomePAM is not discovering new Cas nucleases per se; it's an approach that can be implemented with newly discovered Cas nucleases to determine the PAM profile.

15. L. 226: specify what these variants are and why they were selected.
16. L. 234-237: what is meant when using the words fast and slow? Some rewording is likely needed.
17. L. 279-281: shifting the location of the PAM library represents the exception rather than the rule across PAM determination approaches. This was also done to provide a comprehensive picture of the PAM profile, whereas the authors could have stuck with the larger PAM library with all positions randomized.

Reviewer #3 (Report for the authors (Required)):

In this study, Yu et al. present GenomePAM, a novel method for characterizing protospacer adjacent motifs (PAMs) directly in mammalian cells. GenomePAM utilizes repetitive sequences in the human genome, bypassing the need for protein purification or synthetic oligos. By employing a 20-nt protospacer that occurs around 16,930 times in a human diploid cell, the method allows for precise PAM characterization. The study demonstrated that GenomePAM effectively identifies PAM requirements for type II and type V nucleases, including complex PAMs for enzymes such as CjCas9 and the near-PAMless SpRY. Furthermore, GenomePAM proved capable of assessing both the potency and fidelity of various Cas nucleases by comparing on-target and off-target cleavage activities across thousands of sites. The method also uncovered differences in genome-wide chromatin accessibility profiles among different cell types, providing insights into CRISPR-Cas targeting dynamics. Overall, I am very convinced by results, and GenomePAM as one of the first true mammalian assays for PAM determination. While I think the paper may not reach the novelty threshold for Nature Biomedical Engineering, as PAM assays are well established (HT-PAMDA, PAM-SCANR, etc.), this is a strong study that should eventually be published.

I would like to suggest some edits to make the manuscript stronger:

1. One of the only other mammalian cell-based assays for PAM determination is PAM-DOSE: <https://www.sciencedirect.com/science/article/pii/S2045976919300173>, which involves a very clever fluorescent based strategy to label and sort cells based on PAM recognition. For Type II Cas9s characterized, at least, it would be imperative to do a size-by-side analysis with GenomePAM, PAM-DOSE, and also HT-PAMDA, so as to resolve differences between the different methods.
2. In the novel Cas discovery section, the authors identify a new Cas12 that they decide to characterize with GenomePAM. Overall, this was very well done. However, very little description on why this Cas was chosen was provided. The authors should conduct bioinformatic-analysis of the PAM preference of this Cas12, and check the agreement with the GenomePAM profile. I also ask the authors to do a novel Cas9 enzyme for analysis as well, replicating the process they did for the Cas12.
3. A strong use-case for GenomePAM would also be as a tool to evaluate mutational profiles of Cas enzymes, and I think it is necessary to show this. The authors should apply a set of mutations to the PAM-interacting domain of either an existing or a new Cas9, and then evaluate the PAM profile of these mutants. This can theoretically also be done in a library format.
4. Are there general criteria for picking a repetitive sequence, like Rep-1 or Rep-2, or can Rep-1/2 be used for any Cas? The authors should discuss this in more detail or provide additional datapoints on different repetitive sequences to target.

Minor Edits

1. I really prefer the PAM plotting style from HT-PAMDA. The current graphs are difficult to read and the axis labels are too small. Please plot the PAM profiles from the HT-PAMDA GitHub.
2. The authors should select a single convention in the manuscript: Genome-PAM or GenomePAM. I prefer the second.

Version 1:

Decision Letter:

Dear Zongli,

Thank you for your revised manuscript, "Mammalian genome repeats and AlphaFold 3 direct PAM characterization and engineering of CRISPR-Cas nucleases", which has been seen by the original reviewers. In their reports, which you will find at the end of this message, you will see that the reviewers acknowledge the improvements to the work and that Reviewer #2 raises a few additional technical criticisms that I am hoping you will be able to address. In particular, please ensure that the manuscript discusses the degree of implementation complexity for GenomePAM.

As before, when you are ready to resubmit your manuscript, please upload the revised files, a point-by-point rebuttal to the comments from all reviewers, the [reporting summary](https://www.nature.com/authors/policies/ReportingSummary.pdf), and a cover letter that explains the main improvements included in the revision and responds to any points highlighted in this decision.

As a reminder, please follow the following recommendations:

- * Clearly highlight any amendments to the text and figures to help the reviewers and editors find and understand the changes (yet keep in mind that excessive marking can hinder readability).
- * If you and your co-authors disagree with a criticism, provide the arguments to the reviewer (optionally, indicate the relevant points in the cover letter).
- * If a criticism or suggestion is not addressed, please indicate so in the rebuttal to the reviewer comments and explain the reason(s).
- * Consider including responses to any criticisms raised by more than one reviewer at the beginning of the rebuttal, in a section addressed to all reviewers.
- * The rebuttal should include the reviewer comments in point-by-point format (please note that we provide all reviewers will the reports as they appear at the end of this message).
- * Provide the rebuttal to the reviewer comments and the cover letter as separate files.

We look forward to receive a further revised version of the work. Please do not hesitate to contact me should you have any questions.

Best wishes,

Pep

Pep Pàmies

Chief Editor, Nature Biomedical Engineering

Reviewer #1 (Report for the authors (Required)):

Thanks to the authors for their efforts in addressing my and other reviewers' questions and also for performing some new experiments. Although I still think that the heterogeneity of endogenous sequences leads to potential minor bias in statistics, I have to agree that the method described in this paper is an alternative assay for PAM identification and evaluation. I have no further comments.

Reviewer #2 (Report for the authors (Required)):

The authors have taken substantive steps to address the reviewers' comments, including assessing cell viability, applying GenomePAM to multiple repeat sequences, subjecting a novel Cas9 to GenomePAM, and engineering the same Cas9 to relax PAM preferences. At the same time, there are still some persisting issues to resolve voiced by multiple reviewers:

1. Despite the assurances of the authors, GenomePAM remains a challenging assay to implement for the first time due to the use of GUIDE-seq and AMP-seq, while the results provided for GenomePAM still parallel those from in vitro or bacterial assays. The authors point out one example with CjCas9 where there was some discrepancy between the identified PAMs and validated PAMs in human cells; however, I see this as an exception rather than the rule, and performing follow-on validation in human cells will still be much faster (and cheaper) than completing GenomePAM from start to end. In addition, GenomePAM would likely require some validation in human cells as well since the screen involves one or two targets, even if there would be more confidence given that GenomePAM was conducted in human cells. If GUIDE-seq was much easier to implement, I for one would be excited to adopt it.

2. The authors still hold that larger PAM libraries get progressively more challenging to manage in vitro unlike in GenomePAM, yet there are concrete arguments why this is not the case. First, in vitro cleavage assays can still manage much greater PAM libraries without any scaling issues. For instance, 1 pmol of a DNA library—a small amount to add to an in vitro reaction—contains ~10¹⁷ molecules that could capture a theoretical 20N library. By using a positive screen that pulls out cleaved sequences (e.g., based on ligation to cut DNA; PMID = 26585795), any PAMs could be readily extracted. The libraries can also be specifically designed, such as combining 6N libraries placed different distances from the target site. The authors hold that that every cell offers a sequencing opportunity, yet the DNA is pooled, and there are still a fixed number of variable sequences present in the library. Thus, it's encouraging that GenomePAM presents enough sequence diversity to extract PAMs; however, the argument that GenomePAM overcomes perceived issues with in vitro assays for PAM determination still rings hollow.

3. The authors still rely on back-of-envelope calculations and sequence logos to convey the diversity of potential PAM sequences. There is likely more diversity at some positions than others, which could complicate PAM determination for some Cas nucleases. A deeper analysis of the sequences—and particularly those that are readily accessible—across the entire region is needed. I would recommend using a three- to five-base window to evaluate whether all possible sequences are present and in what proportion, as this spread should capture the spread of known PAMs. My expectation is that some windows will contain far better sequence coverage than others, which could impact the ability to extract PAMs depending on the nuclease. However, I would be delighted if the authors proved otherwise.

4. One reviewer raised the point about cleavage of sites across the genome leading to cell death. Here, the authors provided reasonable evidence in their rebuttal that the cells survive long enough to allow GUIDE-seq, even if the cells eventually succumb to the massive number of double-stranded DNA breaks. At the same time, the concept of creating extensive double-stranded DNA breaks has been used as a strategy to kill cells harboring such repeats (PMID = 26585795). I recommend noting this strategy when the concept of GenomePAM is first introduced so it's clear the authors are taking what could be considered a surprising (but successful) approach.

Reviewer #3 (Report for the authors (Required)):

The authors have satisfactorily addressed all of my concerns. I don't think, however, that AlphaFold3 should be listed in the title, which is not the focus of the study. I request that the authors remove that from the title and focus on the Mammalian genome repeats story of GenomePAM.

The code is organized well using previously established methods.

Version 2:

Decision Letter:

Dear Zongli,

Thank you for your patience in waiting for the latest feedback from Reviewer #2 on your revised manuscript, "Mammalian genome repeats direct PAM characterization and engineering of CRISPR-Cas nucleases". Please see below. Would you be able to provide the code and its output for the iterative seed-extension algorithm?

As before, when you are ready to resubmit your manuscript, please upload the revised files, and a reply to the reviewer's concern.

Best wishes,

Pep

Pep Pàmies

Chief Editor, <http://www.nature.com/nbme> > *Nature Biomedical Engineering*

Reviewer #2 (Report for the authors (Required)):

The authors identified and cited the appropriate/intended reference to genome shredding and sufficiently described pro's and con's of in vitro methodologies.

The authors also described developing an iterative seed-extension algorithm to better explore the spread of sequences flanking targeted repetitive elements. However, I could not find this algorithm or the ensuing output within the submitted materials; at most, there was the added highlighted text about an upstream and downstream 10-bp window in the Methods section on line 387 but no added descriptions in the Results, Figures or Supplemental Information. This analysis remains critical for potential users to gauge which PAMs would be difficult to extract.

Version 3:

Decision Letter:

Dear Zongli,

Thank you for the latest version of your revised manuscript, "GenomePAM directs PAM characterization and engineering of CRISPR-Cas nucleases using mammalian genome repeats". Having consulted with Reviewer #2 (whose comments you will find at the end of this message), I am pleased to write that we shall be happy to publish the manuscript in *Nature Biomedical Engineering*.

We will be performing detailed checks on your manuscript, and in due course will send you a checklist detailing our editorial and formatting requirements. You will need to follow these instructions before you upload the final manuscript files.

Best wishes,

Pep

Pep Pàmies

Chief Editor, Nature Biomedical Engineering

Reviewer #2 (Report for the authors (Required)):

The authors have clarified the availability of the analysis code, and the Github site looks sufficiently usable to enable others to implement GenomePAM. I have no other comments and applaud the authors for their efforts to make this novel methodology accessible and well demonstrated.

Code availability:

I haven't dealt directly with original code in many years, so I wouldn't be able to provide sufficient insights.

Version 4:

Decision Letter:

Dear Dr Zheng,

I am happy to inform you that your manuscript, "GenomePAM directs PAM characterization and engineering of CRISPR-Cas nucleases using mammalian genome repeats", has now been accepted for publication in *Nature Biomedical Engineering*.

Over the next few weeks, the figures will be checked for production quality, the text edited to ensure that it conforms to house style, and the manuscript typeset.

Our Articles are published about 40 days after the acceptance date (we recommend that you inform your institutional press office of this timeframe), and you will be notified of the actual publication date a few days in advance. Articles can be published any working day of the week, and are pushed live shortly after 10 am London time.

Publishing agreement. You will be asked to digitally sign a publishing agreement (grant of rights). After the signed publishing agreement has been received, the proofs of the article will be sent to you for review. If you have any queries during the production process, or you cannot meet the requested deadline for returning the proofs, please contact rjsproduction@springernature.com.

Nature Biomedical Engineering is a Transformative Journal. Authors may publish their research with us through the traditional subscription access route, or make their paper immediately open access through payment of an article-processing charge. More information about publication options is available.

You may need to take specific actions to comply with funder and institutional open-access mandates. If the work described in the accepted manuscript is supported by a funder that requires immediate open access (as outlined, for example, by Plan S) and your manuscript was originally submitted on or after January 1st 2021, then you should select the gold OA route. Authors selecting subscription publication will need to accept our standard licensing terms (including our self-archiving policies), and these will supersede any other terms that the author or any third party may assert apply to any version of the manuscript.

Acceptance of your manuscript is conditional on agreement, by all authors, with both our [media embargo](http://www.nature.com/authors/policies/embargo.html) and [confidentiality and pre-publicity](http://www.nature.com/authors/policies/confidentiality.html) policies. In particular, you may arrange your own publicity of the Article (for instance, through your institutional press office), as long as you ensure that journalists strictly adhere to the media embargo.

To assist you in disseminating the work, as soon as the Article is published you will be able to take advantage of the Springer Nature [SharedIt](https://www.springernature.com/gp/researchers/sharedit) initiative to [generate a unique shareable link to the Article](http://authors.springernature.com/share) that will allow anyone (with or without a subscription) to read it. Recipients of the link who are subscribers will also be able to download and print the PDF.

Thank you for having submitted this work to *Nature Biomedical Engineering*.

Best wishes,

Barbara Cheifet
Editor
Nature Biomedical Engineering

Reviewer #1 (Report for the authors (Required)):

PAM is crucial for CRISPR tools to locate target sites. Various CRISPR tools have evolved to utilize different PAMs, making the identification of accurate PAMs for new CRISPR tools essential. Several strategies exist for identifying PAMs, and in this paper, Yu et al. propose a new and smart strategy named genomePAM, which utilizes endogenous genomic repeat sequences. This method is advantageous as it reduces the need to design diverse PAM-associated sequences. However, several important concerns must be addressed before it can become a very useful tool:

- We thank the reviewer for the appreciation of the advantageous aspects of our new method.

Main concerns:

1. The genomePAM method integrates GUIDE-seq, which requires skills to handle and is more complex than the target sequencing methods widely used in other PAM-identification techniques. The authors should discuss this point in the Discussion section. Moreover, while AMP-seq is mentioned as a potential alternative to GUIDE-seq, no AMP-seq data is provided. The authors should test and present the AMP-seq-involved version.

- GUIDE-seq indeed requires more skills to perform compared with conventional PCR amplicon sequencing. We have now added in the Discussion: “Compared with other PAM identification methods using regular PCR amplicon sequencing, GenomePAM uses GUIDE-seq and thus requires relatively more skills to perform”. We intended to refer to AMP-seq as the enrichment method for dsODN-integrated fragments. We have revised the sentence and spell out the abbreviations in Introduction as “...such as the Genome-wide Unbiased Identification of Double Strand Breaks (DSBs) Enabled by sequencing (GUIDE-seq) that enriches double strand oligodeoxynucleotide (dsODN)-integrated fragments by Anchor Multiplex PCR sequencing (AMP-seq)”. GUIDE-seq is being continuously developed and optimized by us (in The Tsai Lab) to ease its adoption by more labs including streamlining it to use Tn5 tagmentation and a single PCR step for library preparation. We believe GUIDE-seq would become a more and more used tool in both research and therapeutic settings.

2. Several factors make data normalization complicated and may affect the accuracy of identified PAMs. The authors should address these challenges with new data or discuss them in more details.

2.1 As shown in Figure 6, different target sites exhibit varying levels of chromatin accessibility. Therefore, each site should have a different weight when calculating the consensus sequence of the PAM. Did the authors use some way to normalize the accessibility?

- Chromatin accessibility does affect the Cas nuclease activity, as we showed at the genome-wide scale (now **Fig. 7**). The strong correlations between GenomePAM and each the three studies, a method using large-scale synthetic oligos (Kim et al, Nat. Biotech., 2020; **Fig. 2k**), HT-PAMDA (**Fig. 2l**), and PAM-DOSE (**Fig. 2m**), on PAM specificities including those specificities of non-canonical PAMs, suggest that chromatin accessibility does not affect the final GenomePAM results. One reason for the unbiased GenomePAM results is that there are still large numbers of accessible loci in every single cell and, typically, hundreds of thousands of cells are used in one experiment. However, we do share the concern from the reviewer and have added in the Discussion: “The GenomePAM assay is minimally biased by chromatin accessibility, likely due to the large number of accessible perfect-match targets in each cell”. Considering different genomic background compositions of potential PAMs, we have further

developed an iterative seed-extension algorithm to report such genome background compositions, along with the corresponding PAMs (please see our reply to Question 2.3).

2.2 Editing Alu repeats may affect cell activity, potentially impacting cell proliferation or viability. This is a concern for all endogenous sites. The authors should consider whether there are sites in non-functional regions or elements that could mitigate this issue.

- We have conducted new experiments to assess cell viability under 4 different conditions and on two different cell lines. The results showed largely similar cell viabilities across different transfection conditions at 0 h, 24 h and 48 h after Lipofectamine 3000 transfection: (1) SpCas9 plasmid + Rep-1 sgRNA plasmid + dsODN; (2) SpCas9 plasmid + Rep-1 sgRNA plasmid; (3) SpCas9 plasmid + none targeting sgRNA plasmid + dsODN; and (4) Lipofectamine 3000 only, in HEK293T cells (now **Fig. S2a**) and HepG2 cells (**Fig. S2b**). One previous study evaluated the effect of CRISPR-Cas9 editing on cell toxicity in the repetitive element LINE1 (Smith CJ, *NAR*, 2020, PMID: 32315033). Three days after transfection of two LINE1 targeting guides, they observed on average 339 to 2271 edits per haploid genome. Although the edits disappeared on day nine (due to toxicity), they still observed edits on day five. Both our new experiments and the previous study support the notion that transfecting enough cells in aggregate and on day three post transfection can provide enough signal for PAM characterization. Also, as mentioned above in our answer to the previous question, the strong correlations between GenomePAM and each of the three previous PAM determining methods suggest that GenomePAM is accurate during the typical short timeframe of the experiment.

2.3 From Figure 1C, it appears that the spacer sequences of used repeats contain diverse mismatches, which may affect the cleavage frequency of different sites. The authors need to explain how they normalize this variability.

- We calculate PAMs derived from a group of perfect-match and a group of mismatch spacer sequences separately (e.g., now **Fig. 1e**). The two group PAMs appear very similar across different Cas nucleases (**Figs. 2a,b,c, 3a,b,c,g, and 4b,g**) and in three other cells than HEK293T: HepG2, Huh7 and HeLa (**Fig. S3**). Furthermore, we have also developed an algorithm GenomePAM Table that accounts for genomic background composition (**Fig. 1f**), please see the *GenomePAM Table* section in Materials and Methods:

To identify enriched PAM motifs over genomic background, we developed a GenomePAM algorithm (available on GitHub: <https://github.com/Zheng-NGS-Lab/genomePAM>) that involves the computational steps: (1) Identify the most significantly enriched single-base motif: We define the edited value as the sum of GUIDE-seq detected genomic site numbers and GUIDE-seq read counts, with the latter linearly scaled to match the range of the former. The maximum value equals the highest number of genomic sites considered for all combinatorial potential motifs. Within the same motif window, a Chi-square test is used to compare the edited value against the corresponding genomic background counts among all motifs. (2) Extend from the position identified in Step 1 bidirectionally: Extend one base towards the 5' end or one base towards the 3' end and calculate the new edited values. Between the two extensions, the one with higher statistical significance is recorded and used for the next round of extension. (3) Repeat Step 2: Continue extending in both directions until the ends of candidate bases are reached. Record all significant motifs without limiting motif length. (4) Report enriched motifs: Report the enriched motifs along with the percentages of corresponding genomic sites edited, retaining only those motifs with increasing percentages from each iteration step. (**Figs. 2g,h,i, 3g,h,i, 4d,i, and 5d,g**).

For example, SpCas9 and SaCas9 GenomePAM Tables:

Num of perfect match reads: 78207				
PAM (3')	Genome	Edits	Percent (%)	P-value
..G.....	1681	1103	65.6	0.0e+00
..GG.....	477	449	94.4	2.0e-104

Num of perfect match reads: 12352				
PAM (3')	Genome	Edits	Percent (%)	P-value
..G.....	1681	750	44.6	0.0e+00
..GG.....	298	185	62.1	3.9e-130
..GA.....	702	519	73.9	3.5e-130
..GCA....	80	64	80.0	5.0e-23
..GGG....	66	58	87.9	5.0e-23
..GAG....	134	123	91.8	5.0e-23
..GAA....	268	250	93.3	5.0e-23
..GGAT...	30	29	96.7	2.4e-02
..GAGT...	51	50	98.0	2.4e-02

2.4 Whether it's necessary to use different repeat spacer sequences to verify the identified PAMs to exclude the impact of single spacer sequence?

- We have done new experiments using different repeats as spacer sequences, Rep-1, Rep-2, and Rep-3. The results on various Cas nucleases, including type II and type V, all showed largely similar PAMs using different repeat sequences (**Supplementary Fig. S4, S6, S7**). We agree that this question raises a valid concern, also shared by other reviewers, and we have added in Discussion: "However, a new Cas nuclease might have a scaffold sequence that interferes with the repeats, potentially forming strong secondary structures and affecting GenomePAM results. We recommend using at least two different repeats as GenomePAM spacers for novel Cas nucleases. Another possibility is to combine different repetitive sequences in one experiment, although we have not tested this ourselves yet. In such a case, bioinformatic analysis would need to use one repetitive sequence at a time and repeat the analysis for all sequences".

3. When comparing the fidelity of several high-fidelity Cas9 variants, the variation in PAM and spacer sequences across different sites makes the comparison of cleavage activity at off-target sites complicated. The authors should analyze and show the diversity of combined spacer sequences and PAMs to ensure sufficient diversity for this experiment.

- In fact, an unique strength of GenomePAM is its extremely high capacity in analysing large numbers of on- and off-target sequences across mismatch spacers and non-canonical PAMs simultaneously, instead of conventionally using a handful of endogenous sites (typically 10 to 20) to evaluate the potency and specificity without considering non-canonical PAM recognitions. Besides the diversity of PAM in our selected repeat sequences that enables GenomePAM in characterizing various Cas nucleases, we have done new calculations to demonstrate its capabilities in providing extremely rich sequence diversity for off-target analysis, as followed.

Because Cas nucleases can often tolerate a few bases of mismatches (off-targets), we conduct new calculations for the numbers of 20-nt sequences containing up to 4 bases (including budge) mismatches in one human haploid genome:

For Rep-1, these numbers were 48,207, 206,767, 579,336, and 1,350,488, respectively, and > 2 million in total, in the human genome (hg38). Thus, using Rep-1 or Rep-1RC as the protospacer, there are potentially > 4 million targets in one single human diploid cell.

For the 1,350,488 spacers with 4 bases mismatches, the base compositions from positions 1 to 20 showed highly divers base compositions (visualized below). Among all the mismatches in all 20 positions, the lowest number of mismatch base is G at position 20, which occurs 17,128 times in the human genome.

The number of unique sequences in the 1,350,488 spacers with 4 bases mismatches compared with Rep-1 is 137,843 (base compositions visualized below). Among them, the lowest number of mismatch base is T at position 20, which is in 4,121 unique sequences.

In this study, we have focused on comparing variants of the same Cas9 ortholog, in this case within SpCas9 variants. These SapCas9 variants were designed and engineered to improve fidelity, and all of them recognize the canonical NGG PAM of SpCas9. Nevertheless, we have conducted new experiments using a different spacer sequence – Rep-3 – that occurs ~ 5,856 times in one human haploid genome and flanked by nearly random sequences (**Supplementary Fig. S1**). The fidelity results using Rep-3 are comparable with the results using Rep-1 (now **Fig. 6** vs **Fig. S8**). Thus, combining the two results, from Rep-1 and Rep-3, would give similar results. The results are consistent with the literatures.

Minor points:

1. The names of several methods are abbreviated in the Introduction (HT-PAMDA, GUIDE-seq, AMP-seq). The full names should be provided at their first occurrence.

- Thanks. We have now revised accordingly in the Introduction.

2. Schematics are needed for all CRISPR tools to explain the position of PAMs relative to the spacer sequence. This is particularly important because the position of PAMs varies, as seen with CjCas9.

- Thanks for the suggestion. We have now added a schematic to all relevant figures in **Figs. 1b, 2a,b,c, 3a,b,c, 4b,g, and 5b,e.**

Reviewer #2 (Report for the authors (Required)):

In this work, Yu and coworkers devise a novel approach to characterize CRISPR PAMs and off-targets using repetitive sequences in the human genome. Existing approaches for PAM determination involve large libraries of randomized sequences and are normally performed outside of human cells. The authors' approach, termed GenomePAM, targets known repetitive sequences with highly variable flanking ends. After targeting this site with a Cas nuclease, the flanking sequences that allowed DNA cleavage are determined based on existing approaches such as GUIDE-seq or AMP-seq that integrate a DNA barcode at a cut site and sequence the surrounding region. The authors validate GenomePAM using six established Cas nucleases, and they apply the approach to a previously uncharacterized Cas12a nuclease. The authors next take advantage of the repetitive elements containing some sequence variations, offering a simpler screen to gauge the propensity for off-targeting. Finally, by comparing cleaved sites across cell lines, the authors identify cell line-specific sites presumably affected by the chromatin state.

Within the large number of PAM determination approaches that already exist, GenomePAM is a unique and inventive entry differentiated by its ability to rely on existing sequence conservation in human cells and its ability to determine PAMs and off-target propensities in the very environment in which the nucleases would be applied. At the same time, there are notable limitations to the approach, and the general work lacks depth to argue for GenomePAM over existing methodologies. The application of GenomePAM to a new Cas nuclease was helpful, although the nuclease in itself was not a notable addition to the massive existing set of Cas nucleases.

- We thank the reviewer for recognizing that our method is novel, unique and inventive, as well as for the construction comments on limitations for improvements.

Major comments:

1. GenomePAM does come with important limitations that should be addressed by the authors. Specifically, the PAM library is much smaller than what would be used in all other assays, this library is further reduced by inaccessible sites (e.g., in heterochromatin), and it requires Guide-Seq or AMP-seq that are much more complicated to perform and analyze than traditional amplicon-based sequencing used with all other methodologies.

- The size of PAM library is indeed the key consideration when we started conceiving the new method. After extensive computational analyses, we identified several repeats that are flanked by highly diverse sequences and appear to be ideal for PAM analysis, as we describe in Results – Method design and we have now listed some with comments (**Supplementary Fig. S1**). Among them, Rep-1 occurs 8,471 times in a human haploid genome (hg38). For a typical PAM recognition motif ranging from 1 to 4 bases, the average numbers of PAM motifs with perfect match Rep-1 as their spacer would be $\sim 2,117$ ($= 8,471 / (4^1)$) to 33 ($= 8,471 / (4^4)$), respectively, in one human genome. Thus, Rep-1 provides sufficient redundancy in every single cell for most Cas nucleases. In fact, synthetic oligos can become a rather large library as the length of PAM region increases. Previous methods had attempted, in exceptional cases, using sequential rounds of experiments with progressively lengthened PAM candidate sequences to characterize long PAM. In contrast, a single one-round experiment based on GenomePAM, conducted directly in human cells without prior protein purification and without introducing a library of synthetic oligos, was able to precisely identify the PAM preference of CjCas9 being NNNRYAC.

Furthermore, because Cas nucleases can often tolerate a few bases of mismatches (off-targets), we have done new calculations for the numbers of 20-nt sequences containing up to 4 bases (including budge) mismatches in one human haploid genome:

For Rep-1, these numbers were 48,207, 206,767, 579,336, and 1,350,488, respectively, and > 2 million in total, in the human genome (hg38). Thus, using Rep-1 or Rep-1RC as the protospacer, there are potentially > 4 million targets in one single human diploid cell.

This result, along with the fact that typically hundreds of thousands of cells are used in one experiment, has motivated us to use 10 bases as the PAM candidate region, which remains highly random:

- Regarding chromatin accessibility, please also refer to our reply to Reviewer 1 Question 2.1. The strong correlations on PAM results, including canonical and non-canonical PAMs, between GenomePAM and each of the three previously established and unbiased methods demonstrated that GenomePAM provides unbiased results.

- Regarding GUIDE-seq and AMP-seq, please refer to our reply to Reviewer 1 Question 1.

2. An initially attractive feature of the methodology is that it's entirely performed in human cells, allowing the authors to gain context-specific insights free of potential biases in other contexts (e.g., in vitro assays). However, the extracted insights at best match what was extracted from non-mammalian approaches, suggesting that prior approaches would be sufficient. Providing an example where the mammalian insights make a difference would be incredibly helpful to support the methodology; otherwise, I expected the field continue using much easier approaches that do not rely on complicated sequencing approaches like GUIDE-Seq and AMP-seq.

- Advantages and limitations of various PAM characterisation methods was reviewed previously by us (The Kleinstiver Lab, Walton et al, Nat. Protoc. 2021, Table 1). In the study by Fonfara et al., the prokaryotic in-silico approach initially used for searching CjCas9 PAM showed a 'NNNN[C/A]CA' PAM (see below, Left). The accuracy of in-silico approach is limited by available information in the database. The study continued with an in vitro cleavage assay, using the predicted PAM sequence as the substrate, and showed that CjCas9 PAM was 'NNNNACA' (see below, Middle, column 'PAM').

(From Fonfara et al. [PMID: 24270795]: **Left**, Figure 5A in the original study, showed in-silico bacterial database analysis of PAM for CjCas9; **Middle**, originally Figure 5B, DNA substrates designed for specific PAM verification.; **Right**, originally Figure 5C, in vitro plasmid cleavage assay results on plasmid DNA with the 10-bp protospacer adjacent sequence summarized in 5B).

Later, in another study, Kim et al. (PMID: 28220790) reported that: "...CjCas9 recognized 5'-NNNNACAC-3' or 5'-NNNNRYAC-3' (where R and Y stands for purines and pyrimidines, respectively) as PAMs in vitro" and that "To confirm the PAM specificity in human cells, we performed cell-based reporter assays by co-transfecting plasmids encoding CjCas9 and its sgRNA and reporter plasmids containing the target site with variable PAM sequences between RFP and GFP sequences... These reporter assays showed that CjCas9 cleaved target sites containing 5'-NNNNRYAC-3' PAM sequences in HEK 293 cells.". The HEK 293 cell result was consistent with our GenomePAM result.

These results suggest that in vitro assays could show different results than human cell-based assay. Compared with the 'first in vitro screening, then human cell verification' approach, a direct PAM screening assay in human cell would be desirable. Also, GenomePAM allows for evaluating the fraction of semi-canonical to canonical PAM activity in mammalian cellular context, without being affected by cleavage kinetics and limited control of reaction conditions in bacteria or bacterial lysate.

3. The authors also argue that traditional randomized libraries have strong biases or are difficult to maintain. However, this issue is a relatively minor one, particularly for PAM screening. In addition, the set of available sequences flanking the repetitive elements likely also has strong biases on top of covering only a small percentage of possible sequences. Here, a direct comparison to a complete PAM library would be helpful.

- Regarding library size, as in our reply to the first question, for a typical PAM recognition motif ranging from 1 to 4 bases, the average numbers of any particular PAM motif with perfect match Rep-1 would be $\sim 2,117 (= 8,471 / (4^1))$ to $33 (= 8,471 / (4^4))$, respectively, in one human genome. Thus, Rep-1 provides sufficient redundancy in EVERY SINGLE cell for most Cas nucleases. For longer PAMs, the numbers of any PAM motif of lengths 5 and 6 bases, would be 8 and 2, respectively, in one human genome. Yet, typically, hundreds of thousands of cells are used in one experiment. In fact, synthetic oligos can become a rather large library as the length of PAM region increases. Previous methods had attempted, in exceptional cases, using sequential rounds of experiments with progressively lengthened PAM candidate sequences to characterize long PAM. Furthermore, because Cas nucleases can often tolerate a few bases of mismatches (off-targets), we have done new calculations for the numbers of 20-nt sequences containing up to 4 bases (including budge) mismatches in one human haploid genome:

For Rep-1, these numbers were 48,207, 206,767, 579,336, and 1,350,488, respectively, and > 2 million in total, in the human genome (hg38). Thus, using Rep-1 or Rep-1RC as the protospacer, there are potentially > 4 million targets in one single human diploid cell.

The result on \sim millions of targets in EVERY SINGLE cell, along with the fact that typically hundreds of thousands of cells are used in one experiment, has motivated us to use 10 bases as the PAM candidate region, which remains highly random (see above in our reply to Question 1).

- We have reported direct comparisons to each of the three previously established assays. The analyses showed high correlations (**Fig. 2k,l**, vs two assays: $R = 0.96$, $P < 1 \times 10^{-100}$; **Fig. 2m**, vs PAM-DOSE: $R = 0.92$, $P < 2.6 \times 10^{-27}$).

4. For all off-target analyses, I had difficulty understanding what is defined as an off-target and whether additional insights can be drawn beyond a given nuclease being more or less

potent or specific. For instance, is it clear where mismatches are accommodated across the guide? And how would these biases change when starting with an entirely distinct target (e.g., Rep-2)? Finally, does the screen provide at least equivalent information compared to existing off-target screens that can inform off-target prediction?

- The GenomePAM method utilizes GUIDE-seq in off-target validation, a well-established assay for in-cell identification of off-target effects. Indeed, Reviewer 1 raised a similar question. Please see our reply to Reviewer 1 Question 2 where we have done new experiments using a new Rep-3 that occurs ~ 5,856 times in one human haploid genome, and flanked by nearly random sequences that are suitable for use as a spacer for GenomePAM analysis.

5. In general, analyses throughout the work could use much more depth. For instance, how many different repetitive sequences were identified, what is their frequency, distribution, and associated PAM profile? What biases or unexpected cleaved sequences were identified for each nuclease, and what fraction of sites that should have been cleaved but weren't (e.g., due to chromatin accessibility)? Finally, how do the accessible and inaccessible sites align with heterochromatin in these cell lines (since chromatin is never directly mapped)?

- We have now added a list of 20-nt repetitive spacers in human genome (hg38), their occurrences, and with diverse flanking sequences (**Supplementary Fig. S1**). Among them, we have added experiments using the new Rep-3. Genome-wide distribution of Rep-1 is visualized by an ideogram (added as **Fig. 1a**).

Regarding fraction of sites that should have been cleaved but weren't, for each nuclease, we have added in our GenomePAM algorithm a new output table – GenomePAM Table – reporting the bases that are significantly enriched over genomic background and reported their editing percentages (**Figs. 2g,h,i, 3g,h,i, 4d,i, and 5d,g**). Regarding chromatin accessibility, please refer to our reply to the first question.

Other comments:

6. In describing different existing PAM determination methods, one particularly absent set involve cell-free transcription-translation. These approaches overcome many of the hurdles noted for in vitro and in vivo methods, even if they are not in a mammalian context. At the very least, two important reports of these methods should be incorporated: PMID =

29304331, 35216669.

- Thanks for pointing out the missing references. We have added them in the TXTL assays in the Introduction.

7. What other repetitive sequences were identified? The authors briefly mention Rep-2, although a more systematic presentation is needed. These analyses would be good to include in the SI.

- We have added a list of identified 20-nt spacers, with occurrences in human genome (hg38) and flanking sequence diversity. We also commented that Rep-2 may not be an ideal one for GenomePAM due to uneven bases at positions 9 and 10 (though we initially used Rep-2 for experiment as we started out to analyse an 8-nt PAM); Rep-U1 is unsuitable for GenomePAM due to too simple sequence; and Rep-U2 is unsuitable due to low-diversity flanking sequence (**Supplementary Fig. S1**, please see the Note). We would recommend using Rep-1, Rep-3 and Rep-4 for the GenomePAM analysis.

Supplementary Figure S1. Occurrence and flanking sequence diversity of selected repetitive sequences.

Name	Sequence (5' - 3')	Occurrences in human genome (hg38)	Flanking sequence diversity	Note
Rep-1	GTGAGCCACTGTGCCTGGCC	8,471		For 3' PAM
Rep-1RC	GGCCAGGCACAGTGGCTCAC	8,471		For 5' PAM
Rep-2	GAGCCACCGTGCCTGGCCTC	1,126		Not ideal, uneven bases at positions 9 and 10
Rep-2RC	GAGGCCAGGCACGGTGGCTC	1,126		Not ideal, uneven bases at positions -9 and -10
Rep-3	GTGAGCCACCGCACCTGGCC	5,856		For 3' PAM
Rep-3RC	GGCCAGGTGCGGTGGCTCAC	5,856		For 5' PAM
Rep-4	GTGAGCCACCGCGCCTGGCC	9,089		For 3' PAM
Rep-4RC	GGCCAGGCGCGGTGGCTCAC	9,089		For 5' PAM
Rep-U1	CACACACACACACACACA	505,405		Unsuitable, too simple sequence
Rep-U2	CTCCCAAAGTGCTGGGATTA	356,589		Unsuitable, low-diversity flanking sequence

8. In some places the authors use the term Cpf1, which is now outdated and no longer used. Cas12a should be used instead in line with the unified naming of Cas nucleases.

- Thanks. We have revised accordingly throughout the manuscript.

9. Use prime symbols rather than apostrophes when referring to either end of a nucleic acid.

- Thanks. We have revised accordingly throughout the manuscript.

10. L. 150: In this section, SpCas9 was included in the validation set, yet it was already demonstrated in the prior section. Some rewording is in order.
- Thanks. We have revised the heading as “*Validation of GenomePAM on SaCas9 and FnCas12a*”
11. L. 168: FnCas12a is known to be biased against a T at the -1 position. To what extent was this observed? As this bias is common across Cas12a nucleases, is it also seen with RuCas12a?
- We did observe a lower proportion of T at -1 position than the average of 4 nucleotides, according to our GenomePAM Tables of FnCas12a and RuCas12a. However, the difference of the proportions among the four nucleotides was not statistically significant.
12. L. 187-188: does this refer to repetitive sequences with some deviations? Some clarification is necessary.
- No, they were not sequence deviations. There were descriptions in preceding sentence: “...extended Rep-1 to 21 (5' YGTGAGCCACTGTGCCTGGCC 3', Y is C or T) and 22 (5' GYGTGAGCCACTGTGCCTGGCC 3') bases”.
13. L. 194-195: it's worth acknowledging that many other PAM determination methodologies involve positive screens and thus isn't a unique feature of GenomePAM.
- We agree and have rephrased the sentence as “Being a positive selection method, GenomePAM found that as expected, SpRY exhibited a very minimal PAM requirement...”.
14. L. 200-202: some rewording needed, as GenomePAM is not discovering new Cas nucleases per se; it's an approach that can be implemented with newly discovered Cas nucleases to determine the PAM profile.
- We agree. The sentences are revised to: “*GenomePAM for characterizing novel Cas PAM* ...we sought to demonstrate its utility in PAM identification for novel Cas discovery”.
15. L. 226: specify what these variants are and why they were selected.
- We have added “...seven variants in parallel, six of which were compared in a previous study (SpCas9-HF1, eSpCas9(1.1), HypaCas9, xCas9, and Sniper-SpCas9) and the same group developed a new variant (Sniper2L-SpCas9)...”.
16. L. 234-237: what is meant when using the words fast and slow? Some rewording is likely needed.
- Thanks. We have rephrased the sentences as: “The number of on-target sites identified given the same amount of sequencing data was the highest (the most potent) in WT, followed by Sniper2L-SpCas9, and then comparably in Sniper-SpCas9 and eSpCas9(1.1), and the lowest in SpCas9-HF1, HypaCas9 and xCas9 (**Fig. 6c**). The numbers of off-target sites identified given the same amount of data were the lowest (the most specific) in xCas9, HypaCas9, SpCas9-HF1, and eSpCas9(1.1), and then comparably in Sniper-SpCas9 and Sniper2L-SpCas9, and the highest (the least specific) in WT (**Fig. 6d**)”.
17. L. 279-281: shifting the location of the PAM library represents the exception rather than the rule across PAM determination approaches. This was also done to provide a comprehensive picture of the PAM profile, whereas the authors could have stuck with the larger PAM library with all positions randomized.
- We have added in the manuscript “..., in exceptional challenging situations, ...”.

Reviewer #3 (Report for the authors (Required)):

In this study, Yu et al. present GenomePAM, a novel method for characterizing protospacer adjacent motifs (PAMs) directly in mammalian cells. GenomePAM utilizes repetitive sequences in the human genome, bypassing the need for protein purification or synthetic oligos. By employing a 20-nt protospacer that occurs around 16,930 times in a human diploid cell, the method allows for precise PAM characterization. The study demonstrated that GenomePAM effectively identifies PAM requirements for type II and type V nucleases, including complex PAMs for enzymes such as CjCas9 and the near-PAMless SpRY. Furthermore, GenomePAM proved capable of assessing both the potency and fidelity of various Cas nucleases by comparing on-target and off-target cleavage activities across thousands of sites. The method also uncovered differences in genome-wide chromatin accessibility profiles among different cell types, providing insights into CRISPR-Cas targeting dynamics. Overall, I am very convinced by results, and GenomePAM as one of the first true mammalian assays for PAM determination. While I think the paper may not reach the novelty threshold for Nature Biomedical Engineering, as PAM assays are well established (HT-PAMDA, PAM-SCANR, etc.), this is a strong study that should eventually be published.

- We appreciate the reviewer for the recognition of our novel method and strong results.

I would like to suggest some edits to make the manuscript stronger:

1. One of the only other mammalian cell-based assays for PAM determination is PAM-DOSE: <https://www.sciencedirect.com/science/article/pii/S2045976919300173>, which involves a very clever fluorescent based strategy to label and sort cells based on PAM recognition. For Type II Cas9s characterized, at least, it would be imperative to do a size-by-side analysis with GenomePAM, PAM-DOSE, and also HT-PAMDA, so as to resolve differences between the different methods.

- Thanks for the suggestion. We have now compared GenomePAM with PAM-DOSE. The correlation between the two assays on SpCas9 PAM result ($R = 0.92$) (Fig. 2m).

2. In the novel Cas discovery section, the authors identify a new Cas12 that they decide to characterize with GenomePAM. Overall, this was very well done. However, very little description on why this Cas was chosen was provided. The authors should conduct bioinformatic-analysis of the PAM preference of this Cas12, and check the agreement with the GenomePAM profile. I also ask the authors to do a novel Cas9 enzyme for analysis as well, replicating the process they did for the Cas12.

- The reason we chose this Cas was simply because its data was available from the most

recently updated database. We have now added the description: “Using a metagenomics approach (see Methods) to analyze recent data in NCBI Sequence Read Archive (SRA)...”. We have done an in-silico analysis against the massive metagenomic data (Matteo Ciciani, et al., 2022). We established bioinformatic analysis pipeline for PAM prediction. For SpCas9, 36 out of 37 protospacers were identified from the virome database (Moreno Zolfo, et al., 2019; Luis F. Camarillo-Guerrero, et al., 2021; Stephen Nayfach, et al., 2021). The sequence logo derived from 10-bp downstream of the protospacers showed a canonical PAM preference of SpCas9. However, zero protospacer for RuCas12a and only one protospacer for TiCas9 was found in the same virome database.

- For novel Cas9, we have now applied GenomePAM for identifying PAM of a novel Cas9 we recently found from *Tissierella* sp., named TiCas9 (Fig. 4f-j).

TiCas9 clusters closely to SpCas9 and SsCas9, implying that it is a type II-A Cas nuclease (Fig. 4e). GenomePAM analysis revealed that TiCas9 had a NNNACT PAM. We further validated its potencies across 20 endogenous loci in HEK293T cell, which showed up to ~30% editing efficiency using its native gRNA scaffold (Fig. 4h and Supplementary Table 5).

3. A strong use-case for GenomePAM would also be as a tool to evaluate mutational profiles of Cas enzymes, and I think it is necessary to show this. The authors should apply a set of mutations to the PAM-interacting domain of either an existing or a new Cas9, and then evaluate the PAM profile of these mutants. This can theoretically also be done in a library format.

- As suggested, to show such a strong use-case, we set out to engineer the newly identified TiCas9 and used GenomePAM to evaluate its mutational profile. We have added:

GenomePAM facilitates Cas PAM engineering

Engineering Cas PAM preference to expand the targetability represents an attractive strategy for broad applications³⁷. To this end, we questioned whether GenomePAM could facilitate the Cas variant discovery. We showed here using TiCas9 as an example by first applying GenomePAM to profile pooled mutant variants and, upon evidence of altered mixed PAMs, followed by GenomePAM characterization of single-mutant variants (Fig. 5). Because there are many Cas9 nucleases recognizing G/C-rich PAM, we aimed to engineer TiCas9 for recognizing A/T-rich

PAM, namely to relax the C at position 5 of NNNACT. Using AlphaFold 3³⁸, we identified that K1315 was the only residue found to interact with G at position 5 on the complementary strand (Fig. 5a). We constructed an NNK library encoding for all 20 amino acids at position 1315. GenomePAM analysis of the pooled variants showed dramatically altered base compositions at position 5, without affecting positions 4 and 6, in the aggregated PAMs (Fig. 5b,c,d). We then assessed all the 19 a.a. variants individually. The variant K1315Q showed completely no restriction at position 5 (Fig. 5e,f,g), namely an ANT PAM at positions 4 to 6, while other 18 variants showed varied preferences at position 5 (Supplementary Fig. S5). Then, sixteen endogenous sites in *RNF2* harbouring PAM positions 4 to 6 (4 ACT, 4 ATT, 4 AGT, and 4 AAT) were used to validate the variant K1315Q vs WT. The results were consistent with SeqLogo, PCV visualization, and GenomePAM Table (Fig. 5h). Interestingly, GenomePAM Table showed that, for PAM positions 4 to 6, the proportions of genome-wide target sites edited were highest with AGT, followed by ACT, and the lowest with AAT and ATT – largely consistent with the indel percentages at the 16 endogenous sites tested individually (Fig. 5g vs 5h).

Figure 5.

4. Are there general criteria for picking a repetitive sequence, like Rep-1 or Rep-2, or can Rep-1/2 be used for any Cas? The authors should discuss this in more detail or provide additional datapoints on different repetitive sequences to target.

- We think it is possible to combine different repetitive sequences. We recommend using Rep-1, Rep-3, and Rep-4, though we have not tested the combination ourselves yet. In such case, during the bioinformatic analysis, it would need to use one repetitive sequence used at a time and repeat the bioinformatic analysis for all repetitive sequences. We have also added this in Discussion: “Another possibility is to combine different repetitive sequences in one experiment, though we have not tested the combination ourselves yet. In such case, during the bioinformatic analysis, it would need to use one repetitive sequence at a time and repeat the bioinformatic analysis for all repetitive sequences”.

Minor Edits

1. I really prefer the PAM plotting style from HT-PAMDA. The current graphs are difficult to read and the axis labels are too small. Please plot the PAM profiles from the HT-PAMDA GitHub.

- We had changed the original plots and used the HT-PAMDA style in all applicable figures.

2. The authors should select a single convention in the manuscript: Genome-PAM or GenomePAM. I prefer the second.

- Thanks. We have updated accordingly.

Reviewer #1 (Report for the authors (Required)):

Thanks to the authors for their efforts in addressing my and other reviewers' questions and also for performing some new experiments. Although I still think that the heterogeneity of endogenous sequences leads to potential minor bias in statistics, I have to agree that the method described in this paper is an alternative assay for PAM identification and evaluation. I have no further comments.

- We appreciate the reviewer's constructive comments that have helped improved our study.

Reviewer #2 (Report for the authors (Required)):

The authors have taken substantive steps to address the reviewers' comments, including assessing cell viability, applying GenomePAM to multiple repeat sequences, subjecting a novel Cas9 to GenomePAM, and engineering the same Cas9 to relax PAM preferences. At the same time, there are still some persisting issues to resolve voiced by multiple reviewers:

1. Despite the assurances of the authors, GenomePAM remains a challenging assay to implement for the first time due to the use of GUIDE-seq and AMP-seq, while the results provided for GenomePAM still parallel those from in vitro or bacterial assays. The authors point out one example with CjCas9 where there was some discrepancy between the identified PAMs and validated PAMs in human cells; however, I see this as an exception rather than the rule, and performing follow-on validation in human cells will still be much faster (and cheaper) than completing GenomePAM from start to end. In addition, GenomePAM would likely require some validation in human cells as well since the screen involves one or two targets, even if there would be more confidence given that GenomePAM was conducted in human cells. If GUIDE-seq was much easier to implement, I for one would be excited to adopt it.

- We are working on streamlining the GUIDE-seq workflow (in The Tsai Lab), including using Tn5 tagmentation and a single PCR step for library preparation. We will be glad to share the protocol on a preprint server for exchanging comments and feedback as soon as the manuscript is ready. We also added in Discussion: "However, GUIDE-seq has been one of the main methods for assessing CRISPR off-target effects in both research and therapeutic settings (PMID: 34215024 and 33283989). Its workflow is being continuously optimized that we believe would facilitate its easier implementation".

2. The authors still hold that larger PAM libraries get progressively more challenging to manage in vitro unlike in GenomePAM, yet there are concrete arguments why this is not the case. First, in vitro cleavage assays can still manage much greater PAM libraries without any scaling issues. For instance, 1 pmol of a DNA library—a small amount to add to an in vitro reaction—contains ~10¹⁷ molecules that could capture a theoretical 20N library. By using a positive screen that pulls out cleaved sequences (e.g., based on ligation to cut DNA; PMID = 26585795), any PAMs could be readily extracted. The libraries can also be specifically, designed, such as combining 6N libraries placed different distances from the target site. The authors hold that that every cell offers a sequencing opportunity, yet the DNA is pooled, and there are still a fixed number of variable sequences present in the library. Thus, it's encouraging that GenomePAM presents enough sequence diversity to extract PAMs; however, the argument that GenomePAM overcomes perceived issues with in vitro assays for PAM determination still rings hollow.

- We agree with the reviewer that “in vitro cleavage assays can still manage much greater PAM libraries without scaling issues.” We have added this in Introduction: “In vitro cleavage assays have the advantages of able to manage large libraries.”

3. The authors still rely on back-of-envelope calculations and sequence logos to convey the diversity of potential PAM sequences. There is likely more diversity at some positions than others, which could complicate PAM determination for some Cas nucleases. A deeper analysis of the sequences—and particularly those that are readily accessible—across the entire region is needed. I would recommend using a three- to five-base window to evaluate whether all possible sequences are present and in what proportion, as this spread should capture the spread of known PAMs. My expectation is that some windows will contain far better sequence coverage than others, which could impact the ability to extract PAMs depending on the nuclease. However, I would be delighted if the authors proved otherwise.

- We agree that the diversity of potential PAM sequences could complicate PAM determinations. For this very reason, in addition to our initial thorough validations and comparisons of GenomePAM with three other previously established PAM methods including canonical and non-canonical PAMs, we have developed an iterative seed-extension algorithm to calculate objective enrichment (with statistical significances) on potential PAM motifs. Lastly, we also recommend using at least two repeat sequences (such as Rep-1, Rep-3, and Rep-4) as the targets and contain different flanking potential PAM sequences when characterizing new Cas nucleases.

4. One reviewer raised the point about cleavage of sites across the genome leading to cell death. Here, the authors provided reasonable evidence in their rebuttal that the cells survive long enough to allow GUIDE-seq, even if the cells eventually succumb to the massive number of double-stranded DNA breaks. At the same time, the concept of creating extensive double-stranded DNA breaks has been used as a strategy to kill cells harboring such repeats (PMID = 26585795). I recommend noting this strategy when the concept of GenomePAM is first introduced so it's clear the authors are taking what could be considered a surprising (but successful) approach.

- If we understand you correctly, do you mean the ‘cancer genome shredding’ study PMID: 37917583 instead of PMID = 26585795? As you suggest, we have added in Line 131: “Cell toxicity after large numbers of DSBs occur in one cell were reported previously when using CRISPR to target highly repetitive element LINE1 (PMID: 32315033) or unique repeat sequences associated with temozolomide mutational signature (PMID: 37917583)...”

Reviewer #3 (Report for the authors (Required)):

The authors have satisfactorily addressed all of my concerns. I don't think, however, that AlphaFold3 should be listed in the title, which is not the focus of the study. I request that the authors remove that from the title and focus on the Mammalian genome repeats story of GenomePAM.

- We agree and have revised the title as “GenomePAM directs PAM characterization and engineering of CRISPR-Cas nucleases using mammalian genome repeats”.

The code is organized well using previously establish methods.

- Thank you.

Reviewer #2 (Report for the authors (Required)):

The authors identified and cited the appropriate/intended reference to genome shredding and sufficiently described pro's and con's of in vitro methodologies.

The authors also described developing an iterative seed-extension algorithm to better explore the spread of sequences flanking targeted repetitive elements. However, I could not find this algorithm or the ensuing output within the submitted materials; at most, there was the added highlighted text about an upstream and downstream 10-bp window in the Methods section on line 387 but no added descriptions in the Results, Figures or Supplemental Information. This analysis remains critical for potential users to gauge which PAMs would be difficult to extract.

-- Thank you. We previously included our GenomePAM GitHub repository under the '*Code availability*' section. We have now added the repository link already in the '*GenomePAM Table*' section. For quick reference, the iterative seed-extension steps (https://github.com/Zheng-NGS-Lab/genomePAM/blob/main/bin/GenomePAM.v2_240901.R) are described as:

Computation steps:

- # 1. Identify the strongest single base among the PAM bases and its position.
- # 2. Extend PAM bases from Position A one base towards left and one base towards right
- # 3. Calculate the difference among the extended sequences in either direction, Left and Right, respectively and record the two P values.
- # 4. Compare the P values. The more significant one is selected as the enriched PAM and used as the basis for the next round of extension.
- # 5. Repeat Step 2 to 4 until the ends of candidate PAM bases.